# Evaluating the representation of aerosol optical properties by an on-line coupled model over the Iberian Peninsula

Laura Palacios-Peña[1], Rocío Baró[1], Juan L. Guerrero-Rascado[2, 3], Lucas Alados-Arboledas[2, 3], Dominik Brunner [4], Pedro Jiménez-Guerrero*[1]

[1]Department of Physics, Regional Campus of International Excellence "Campus Mare Nostrum", Campus de Espinardo, University of Murcia, Murcia, 30100, Spain
[2]Andalusian Institute for Earth System Research (IISTA-CEAMA), Av. del Mediterráneo, Granada, 18006, Spain
[3]Department Applied Physics, University of Granada, Fuentenueva s/n, Granada, 18006, Spain
[4]Laboratory for Air Pollution/Environmental Technology (EMPA), Swiss Federal Laboratories for Materials Science and Technology, Dübendorf, 8600 Switzerland

*Correspondence to: Pedro Jiménez-Guerrero (pedro.jimenezguerrero@um.es)

**Abstract.** The effects of atmospheric aerosol particles on the Earth's climate mainly depend on their optical, microphysical and chemical properties, which modify the Earth radiative budget. The aerosol radiative effects can be divided into direct and semi-direct effects, produced by the aerosol-radiation interactions (ARI); and indirect effects, produced by aerosol-cloud interactions (ACI). In this sense the objective of this work is to assess whether the inclusion of aerosol radiative feedbacks in the on-line coupled WRF-Chem model improves the modelling outputs over the Iberian Peninsula (IP) and surrounding water areas. For that purpose, the methodology bases on the evaluation of modelled aerosol optical properties under different simulation scenarios. The evaluated data come from two WRF-Chem simulations for the IP differing in the inclusion/no-inclusion of ARI and ACI (NRF/RF simulations). The case studies cover two episodes with different aerosol types over the IP in 2010, namely a Saharan desert dust outbreak and a forest fire episode. The evaluation uses observational data from AERONET stations and MODIS sensor, including aerosol optical depth (AOD) and Ångström exponent (AE). Experimental data of aerosol vertical distribution from the EARLINET Granada station are used for checking the models. The results indicate that for the spatial distribution the best-represented variable is AOD and the largest improvements of including the aerosol radiative feedbacks are found for the vertical distribution. In the case of the dust outbreak, a slight improvement (worsening) is produced over the areas with medium (high/low) levels of AOD (-9%/+12% of improvement) when including the aerosol radiative feedbacks. For the wildfires episode, improvements of AOD representation (up to 11%) over areas further away from emission sources are estimated, which compensates the computational effort of including aerosol feedbacks in the simulations. No evident improvement is observed for the AE representation, whose variability is largely underpredicted by both simulations.

# 1 Introduction

It is nowadays widely recognised that aerosol particles exert a substantial influence on Earth's climate changing the radiative budget (Charlson et al., 1992; Hansen et al., 1997; Ramanathan and Feng, 2009; Boucher et al., 2013, IPCC, 2013, among many others). The principal mechanisms by which aerosols cause these changes are: (1) scattering and absorption solar and terrestrial radiation (aerosol-radiation interactions, ARI) (e.g. Ruckstuhl et al., 2008) and: (2) modification of clouds and precipitation, thereby affecting both radiation and hydrology, or increasing the reflectivity of clouds (aerosol-cloud interactions, ACI) (e.g. Twomey, 1974; Albrecht, 1989; Twomey, 1991). In the first case, light scattering by aerosol particles such as sea salt or desert dust increases the solar radiation reflected by the planet, producing a cooling influence. Light-absorbing aerosols such as black carbon, which are a component of biomass burning, exert a warming influence (e.g. Jacobson, 2001). These radiative influences are quantified as forcings (in W m$^{-2}$), defined as the perturbation to the energy balance of the Atmosphere-Earth system. A warming influence is denoted a positive forcing, and a cooling influence, negative (IPCC, 2013). Generally, modelling tools and observations indicate that anthropogenic aerosols have had a cooling influence on Earth since preindustrial time, with a total ARI+ACI medium-confidence radiative forcing (excluding the effect of absorbing aerosol on snow and ice) of -0.9 (-1.9 to -0.1) W m$^{-2}$ (Boucher et al., 2013). The uncertainty quantification of these aerosol effects on the Earth radiative budget is much higher than for any other climate-forcing agent (IPCC, 2013). This happens because the physical, chemical and optical aerosol properties are highly variable in space and time scales due to the aerosol particles short-lived and non-uniform emissions (Forster et al., 2007).

In order to reduce this uncertainty, the use of models is one of the most powerful tools to understand the different processes affecting the climate system. As aerosol may strongly drive the Earth's climate on global and regional scales, fully-coupled meteorology-climate and chemistry models allow for accounting the climate-chemistry-aerosol-cloud-radiation feedbacks mechanisms between simulated aerosol concentrations and meteorological variables. It is also a promising way to go for future atmospheric simulation systems, leading to a new generation of models for improved meteorological, environmental and chemical weather forecasting (Baklanov et al., 2014).

Europe may be one of the most climatically sensitive world regions (Giorgi, 2006). Within the target domain, the role of aerosol particles may then be even more crucial over such regions as the Mediterranean basin, a crossroad that fuels the mixing of particles from different sources (Papadimas et al., 2012). The Iberian Peninsula (IP), as a good example within the Mediterranean basin, can be affected by high aerosols concentration of different aerosol types. Due to its closeness to the Sahara Desert, the IP is frequently affected by dust outbreaks with large aerosol loads that modulate the aerosol climatology in different areas of this region, especially in Southern Spain (e.g. Toledano et al., 2007; Guerrero-Rascado et al., 2008, 2009; Córdoba-Jabonero et al., 2011; Antón et al., 2012; Pereira et al., 2014) and Portugal (e.g. Wagner et al., 2009; Preißler et al., 2011). On the other hand, the Mediterranean climate, with high summer temperatures and dry soil-air conditions, encourage forest fires episodes over this region (Alados-Arboledas et al., 2010). Both types of emissions give major contributions to particle concentration in the atmosphere, particularly in the warmer season (Elias et al., 2006).

There is a large number of studies assessing the aerosol feedbacks effects over the IP using different remote sensing measurement methods, using devices such as sun photometers (Lyamani et al., 2005, 2006; Toledano et al., 2007; Cachorro et al. 2008; Obregón el at., 2012), nephelometers (Pereira et al., 2008, 2011), lidars (Guerrero-Rascado et al., 2008) or a combination of these (Elias et al., 2006; Córdoba-Jabonero et al., 2011). Other studies using these instruments join satellite measurements to carry out this assessment (Cachorro et al., 2006; Guerrero-Rascado et al., 2009). Even, there are studies that use these different measurements to estimate by a radiative transfer model the aerosol radiative forcing over some regions (Santos et al., 2008; Guerrero-Rascado et al., 2009; Valenzuela et al., 2012) or over the whole IP (Mateos et al. 2014). On the other hand, a number of studies (e.g. Myhre et al., 2007; Myhre et al., 2009) have tried to assess the aerosol feedbacks effects on a global scale, while other works (e.g. Péré et al., 2010; Meij et al., 2012; Curci et al., 2014, among others) have a more regional approach. However, no modelling studies of the aerosol radiative effects have ever been carried out for the IP. According to Randall et al. (2007), the responses of the climate system to aerosols and their effects on the radiative budget of the Earth are the most uncertain climate feedbacks.

Therefore, the objectives of this work are (i) to assess whether the inclusion of aerosol radiative feedbacks in the on-line coupled WRF-Chem model improves the modelling outputs of aerosol optical properties (aerosol optical depth, AOD and Ångström exponent, AE) over the IP and surrounding water areas (seas and ocean) and (ii) to evaluate the representation of aerosol optical properties by this model over the target domain.

## 2 Methodology

In this paper we evaluate the AOD and AE output of different simulations carried out by the WRF-Chem model (Grell et al., 2005) by using observational data provided by several instruments: two ground-based data networks (AERONET and EARLINET) and a sensor on-board a satellite (MODIS). The results of the evaluation of the simulations presented here concerning particulate matter concentrations can be found in Im et al. (2015). Therefore, in this contribution we will focus on the evaluation of aerosol properties. Two different setups of the model have been considered, including/not including aerosol radiative feedbacks in the simulation. According to Boucher et al. (2013), the inclusion of these feedbacks involves a change on the internal energy flows to the Earth system, affecting cloud cover or other components of the climate system such as aerosol particles, and, thereby, altering the global energy budget indirectly.

The evaluation has been performed by using classical statistics according to Willmott et al. (1985), Weil et al. (1992) and Willmott and Matsuura (2005). The individual model-prediction error or bias ($e_i$), the mean bias error (MBE), mean absolute error (MAE) and the correlation coefficient (r) have been calculated. All data needs have pre-processed and bilinearly interpolated to a common working grid. This has a resolution of 0.2° and covers between 35° and 47° north and -15 and 5° east. The grid size consist of 6000 cells and the grid type is a regular lon-lat grid. After the interpolation, modelled data are evaluated against MODIS. The data to compare with AERONET and EARLINET are extracted from the model cell covering the corresponding station coordinates (Fig. 1) following a nearest neighbour approach.

First, in order to evaluate whether the inclusion of aerosol radiative feedbacks in the on-line coupled WRF-Chem model produces significant changes on the studied variables (or changes are just mere signal noise), a surrogate variable, associated to the significance level of the changes modelled (S.L.), is defined (Eq.1). Therefore, high values of S.L. indicate whether the changes between simulations including (and not including) aerosol radiative feedbacks are noticeable with respect to the variability of the signal or not, and therefore, their significance:

$$S.L. = \frac{\frac{1}{n}\sum_{i=1}^{n}\left|x_{i_{NRF}} - x_{i_{RF}}\right|}{S_{NRF}^2} \, x100 \; , \tag{1}$$

where S.L. is the significance level, $x_i$ is the value of the studied variable and $S_{NRF}^2$ is the associated variance for the case not taking into account any aerosol radiative feedbacks (NRF). Moreover, NRF represents the base case and RF is the aerosol radiative feedbacks simulation that includes the ARI+ACI.

Second, to evaluate whether the inclusion of the aerosol radiative feedbacks in the simulations leads to an improvement of the error of the model, the variable Improvement of the MAE is used (Eq. 2):

$$Improvement \; of \; MAE = \frac{1}{n}\sum_{i=1}^{n}\left|e_i\right|_{NRF} - \frac{1}{n}\sum_{i=1}^{n}\left|e_i\right|_{RF} \; , \tag{2}$$

where $|e_i|$ is the absolute error of the simulations.

Finally, to estimate whether the inclusion of the aerosol radiative feedbacks in the simulations produces an improvement of the vertical distribution of aerosols, the normalized improvement of MAE has been calculated (Eq. 3):

$$Nor. \, Improv. \, of \; MAE \; (\%) = \frac{1}{n}\left(\frac{\sum_{i=1}^{n}\left|e_i\right|_{NRF} - \sum_{i=1}^{n}\left|e_i\right|_{RF}}{\sum_{i=1}^{n}\left|e_i\right|_{NRF}}\right) x100 \; , \tag{3}$$

## 2.1 Modelling data: WRF-Chem

The evaluated data comes from regional air quality-climate simulations performed using the WRF-Chem online-coupled meteorology and chemistry model (Grell et al., 2005), version 3.4.1, under the umbrella of the EuMetChem Cost Action ES1004. A detailed description of the simulations can be found in Forkel et al. (2015).

A brief description of the modelling methodology taken from the aforementioned work is described below. The following physics options were applied for both simulations, including (or not) aerosol radiative feedbacks: Rapid Radiative Transfer Method for Global (RRTMG) longwave and shortwave radiation scheme; the Yonsei University (YSU) PBL scheme, the NOAH land-surface model, the Lin microphysics scheme and the updated version of the Grell-Devenyi scheme with radiative feedbacks. Further description of the physics can be found in Grell et al. (2005). According to chemistry options, the followings were applied: the Modal Aerosol Dynamics Model for Europe with the Secondary Organic Aerosol Model (MADE/SORGAM) aerosol scheme; the Regional Acid Deposition Model version 2 (RADM2) gas phase mechanism and the Fast-J photolysis scheme.

For all simulations discussed in this paper the native modelling grid spacing is 23 km (270 by 225 grid cells, Lambert Conformal Conic projection with center at 50N and 12E). The modelling domain covers Europe and a portion of Northern Africa and as well as large areas affected by the Russian forest fires. However, because of the scope of the paper is the IP, only data for a domain covering the IP and surrounding seas and ocean has been used (Fig. 2). In the vertical direction, the atmosphere up 50 hPa is resolved into 33 layers with a higher resolution close to the surface.

Initial and boundary conditions for the meteorological variables were obtained from 3-hourly data with 0.25° resolution (analysis at 00 and 12 UTC and respective forecasts 3/6/9 hours) from the ECMWF operational archive. 3-hourly chemistry boundary conditions for the main trace gases and particulate matter concentrations were available from the ECMWF IFS-MOZART model run from the MACC-II project (Monitoring Atmospheric Composition and Climate-Interim Implementation, Inness et al., 2013) 1.125° spatial resolution. Anthropogenic emissions for the EU domain provided by the TNO (Netherlands Organization for Applied Scientific Research) from a recent update of the TNO MACC emissions inventory (http://www.gmes-atmosphere.eu/; Pouliot et al., 2012, 2014; Kuenen et al., 2014) were applied.

Biomass burning emission data have been calculated from global fire emission data that have been supplied from the integrated monitoring and modelling system for wild-land fires (IS4FIRES) project (Sofiev et al., 2009) with 0.1° x 0.1° spatial resolution. Day and night vertical injection profiles were also provided. WRF-Chem emission species have been calculated by speciation following Andreae and Merlet (2001) and Wiedinmyer et al. (2011). However, no heat release due to the fires was taken into account.

Biogenic emissions are based on the Model of Emissions of Gases and Aerosols from Nature (MEGAN) model (Guenther et al., 2006). MEGAN is on-line coupled with WRF-Chem and makes use of simulated temperature and solar radiation. Moreover,

WRF-Chem predicts online dust emission as a function of the land usage information and the simulated meteorological fields. In this work and following Shaw et al. (2008), dust emission flux ($G$) depends on: an empirical proportionality constant estimated based on regional specific data ($C$); the vegetation mask accounting for vegetation type ($\alpha$); the friction velocity ($u_*$); the threshold friction velocity below which dust emission does not occur ($u_{*t} = 20$ cm s$^{-1}$) following Shaw et al., 2008); and the soil wetness factor accounting for soil moisture ($f_w$).

$$G = \alpha C u_*^4 \left(1 - \frac{f_w u_{*t}}{u_*}\right) \qquad (4)$$

The most important feature to bear in mind for this work is the aerosol module. This aerosol module is based on the modal aerosol MADE (Modal Aerosol Dynamics Model) (Ackermann et al., 1998) which is a modification of the Regional Particulate Model (Binkowski and Shankar, 1995). Here aerosol particles are represented by three log-normal size distributions, corresponding to an Aitken mode (nucleation mode 0.1 μm diameter), and accumulation mode (0.1 – 2 μm), and a coarse mode (> 2 μm) (Forkel et al. 2012). SOA have been incorporated into MADE in the SORGAM (Secondary Organic Aerosol Model) module (Schell et al., 2001).

Aerosol chemical properties and sizes are used to determine aerosol optical properties as a function of wavelength using the method outlined in Fast et al. (2006) and Barnard et al. (2010). In brief, each chemical constituent of the aerosol is associated with a complex index of refraction. The overall refractive index for a given size bin is determined by volume averaging, with Mie theory and summation over all size bins used to determine composite aerosol optical properties. Wet particle diameters
are used in the calculations (Chapman et al. 2009).

The microphysical module consist of the Lin scheme based on Lin et al. (1983) and Rutledge and Hobbs (1984), is a single moment scheme including some modifications, as saturation adjustment following Tao et al. (1989) and ice sedimentation, which is related to the sedimentation of small ice crystal (Mitchell et al., 2008). It includes six classes of hydrometeors: water vapour, cloud water, rain, cloud ice, snow, and graupel (Baró el al. 2015). WRF-Chem model allows to transform the single
into a double moment scheme of the Lin microphysics scheme. This implementation is described in Chapman et al. (2009). Following Ghan et al. (1997), a prognostic treatment of cloud droplet number was added, which treats water vapour and cloud water, rain, cloud ice, snow, and graupel. The autoconversion of cloud droplets to rain droplets depends on droplet number and follows Liu et al. (2005). Droplet-number nucleation and (complete) evaporation rates correspond to the aerosol activation and resuspension rates. Ice nuclei based on predicted particulates are not treated. However, ice clouds are included via the
prescribed ice nuclei distribution following the Lin scheme (Baró et al. 2015).

Finally, the effect of aerosols on incoming solar radiation within WRF-Chem is determined by transferring relevant parameters to the shortwave radiation scheme, representing radiative feedbacks due to aerosol-radiation interactions. The interactions of clouds and incoming solar radiation have been implemented by linking simulated cloud droplet number with the shortwave radiation scheme and with Lin microphysics (Skamarock et al., 2005). Therefore, droplet number will affect both the calculated
droplet mean radius and cloud optical depth when using shortwave radiation scheme, representing radiative feedbacks due to aerosol-clouds interactions. A limitation of WRF-Chem in the treatment of aerosol-cloud interactions is that this couplings are not computed in convective clouds simulated by the cumulus parameterisation (Chapman et al., 2009; Yang et al., 2011; Archer-Nicholls et al., 2016).

Although the modelling domain covers all Europe, for the purpose of this work data from the IP and surrounding areas with a
resolution of 0.2° has been extracted for two important aerosol episodes in 2010. One of these episodes consists of a Saharan desert dust outbreak (from 28 June to 12 July) and a forest fires episode (from 25 July to 7 August). These episodes are selected because they represent two situations with a high load of atmospherics aerosol particles, when the radiative budget can be strongly affected. No volcanic emissions were considered in spite of the Eyjafjallajökull eruption in spring 2010. However, the volcanic plume reached all the IP only in May 2010 (Sicard et al., 2012; Navas-Guzmán et al., 2013), which is out of the
scope of these case studies.

The simulations are run for two different configurations differing in the inclusion/no-inclusion of aerosol radiative feedbacks (ARI+ACI). The base case or NRF simulation, does not take into account any aerosol feedbacks and the RF simulation adds the ARI and ACI to the previous modelling setup. At this point, it should be mentioned that the use of ECMWF operational

archive for meteorological initial and boundary conditions can produce that some of the aerosol feedback may already take into account in the base case (NRF) because of the model assimilation of meteorological observations of the ECMWF.

## 2.2 Observational data

### 2.2.1 Moderate Resolution Image Spectrometer (MODIS)

The satellite data chosen to evaluate the WRF-Chem simulations comes from MODIS (Levy et al., 2005) Level-2 Atmospheric Aerosol Product (MXD04_L2), collection or version 6 (C6) (Levy et al., 2013). The MODIS Aerosol Products monitor the ambient aerosol optical thickness over the oceans globally and over a portion of the continents. Daily Level 2 data have a spatial resolution of a 10x10 km. Two MODIS Aerosol data product files have been selected: MOD04_L2, containing data collected from the Terra platform; and MYD04_L2, containing data collected from the Aqua platform. In this case, the

MXD04_L2 provides full global coverage of aerosol properties from the Dark Target (DT) aerosol retrieval algorithm, which is applied over ocean and dark land (e.g., vegetation) (Levy et al., 2013).

The variables used from MODIS are Aerosol Optical Depth (AOD) and Ångström Exponent (AE).

AOD corresponds with AOD at a wavelength of 550 nm ($AOD_{550}$) for both ocean (best) and land (corrected) with best quality data (Quality Assurance Confidence = 3). The valid range of data is -0.05 to 5.0; that means a permission of small negative

AOD values in order to avoid an arbitrary negative bias at the low $AOD_{550}$ end in long-term statistics. This is because MODIS does not have sensitivity over land to retrieve aerosol to better than $\pm 0.05 + 15$ % under very clean conditions. Negative values of $AOD_{550}$ have been considered as zero in this study. Over ocean the estimated error is -0.02 - 10%, +0.04 + 10% (Levy et al., 2013).

AE stands for AE for wavelengths between 550 and 860 nm ($AE_{550/860}$) over the ocean. The valid range for this variable is -

20 1.0 to 5.0. In Collection 6, the preliminary estimated error for $AE_{550/860}$ is 0.45; pixels with an $AOD_{550} > 0.2$ are expected to have a more accurate $AE_{550/860}$ representation (Levy et al., 2013).

### 2.2.2 Aerosol Robotic Network (AERONET)

The Aerosol Robotic Network (AERONET) collaboration (Holben et al., 1998) provides globally distributed observations of spectral AOD, inversion products, and precipitable water in diverse aerosol regimes. The highest quality data can be found in

Version 2, Level 2.0 (cloud-screened and quality-assured) data products.

The data used from AERONET in this work comes from level 2.0 of AOD at different wavelengths ($AOD_{440}$, $AOD_{675}$, $AOD_{870}$ and $AOD_{1020}$) and AE ($AE_{440/870}$) from stations covering the IP available for the episodes studied (Fig. 1). Typically the total uncertainty for AOD data under cloud-free conditions is $< \pm 0.01$ for $\lambda > 440$ nm and $< \pm 0.02$ for shorter wavelengths (Holben et al., 1998).

### 2.2.3 European Aerosol Research lidar Network (EARLINET)

EARLINET (Pappalardo et al., 2014) is the first aerosol lidar network, established in 2000, with the main goal to provide a comprehensive, quantitative, and statistically significant database for the aerosol distribution on a continental scale. EARLINET data include particle backscatter and extinction coefficient profiles at 355, 532 and 1064 nm. EARLINET data used include backscatter profiles (BSCAT) at 355 and 532 nm (for the dates and times selected, no information is available at 1064 nm). The only station with available data for the studies cases in the IP during the year 2010 is Granada, and therefore is the only station included in this study.

## 3 Results and discussion

### 3.1 Significance level of simulated changes

First, to assess the effect of the inclusion of aerosol radiative feedbacks in the on-line coupled WRF-Chem model on the studied variables, the significance study described in Section 2 has been carried out. During the dust episode (Fig. 2), the inclusion of aerosol radiative feedbacks produces differences with a significance level (defined as the ratio for the NRF-RF differences and the associated variance for the case not taking into account any aerosol radiative feedbacks) for $AOD_{550}$ higher than 60% over the south-western IP. The rest of the domain presents S.L. ratios > 100 % in spite of the high $AOD_{550}$ variance values (above 0.05). In the case of $AE_{440/870}$, the entire domain shows significance levels higher than 100%.

The inclusion of aerosol radiative feedbacks during the simulated fire episode (Fig. 3) produces differences with a S.L. > 100%) for $AOD_{550}$ over most of the domain. Over the area of fire particles emissions, S.L. ranges between 50 and 100% due to the higher absolute changes (> 0.2) than variance values (> 0.05). Similarly, for the dust episode the $AE_{440/870}$ over the entire domain shows significance levels > 100%.

Hence, over most of the domain the changes or differences due to the inclusion of aerosol radiative feedbacks have a high S.L., usually higher than 100%, and therefore the changes modelled are significant with respect to the variability of the studied variables. We can then state that the changes discussed below are caused by the inclusion of the aerosol radiative feedbacks in our simulations, and not to the mere signal noise.

### 3.2 Model output vs. Terra-MODIS data

The results of the comparison between model outputs with MODIS data from Terra platform are shown in Figs. 4-7. The results from Aqua Platform are similar to Terra, and are therefore not shown here (but included in the Supplementary Material). Fig. 4 (a) shows the mean values of $AOD_{550}$ from MODIS for the dust outbreak. In this episode, high levels (above 0.4) over the south and the east of the domain are found, due to the shape of the dust outbreak. On the other hand, for the fires episode (Fig. 5 (a)), the highest levels of MODIS $AOD_{550}$ (> 0.25) are shown over the north of Portugal due to the presence of black

carbon coming from wildfires, and over the south of the domain where a dust intrusion occurred at the end of this episode (> 0.3).

For $AOD_{550}$ over the entire domain, the model outputs present low values of the MBE (represented by Figs. 4 and 5, (c) and (d)) for both NRF and RF simulations. During the dust episode the model underestimates MODIS $AOD_{550}$ (MBE minimum values for NRF and RF simulations, respectively, -0.31 and -0.36) over the locations with important dust loads (high $AOD_{550}$) and overestimates (MBE maximum values 0.32 and 0.31) the low levels of $AOD_{550}$. Although the bias is generally lower during the fires episode, a peak of positive bias (0.47 for both simulations) is evaluated over the Portugal fire area, thus the model overestimates $AOD_{550}$ for biomass burning particles for both model configurations, including or not aerosol radiative feedbacks. However, we should bear in mind that this fact may be conditioned by the MODIS underestimation of $AOD_{550}$ levels for high loads of this type of particles, which has been reported by Chu et al., 2002; Levy et al., 2005 and Remer et al., 2005, among others. On the other hand, generally a too high predicted AOD by the model can be explained by either too much aerosol dry mass present in the model, too large fraction of small particles for a given mass, or due to an excess of water associated with the aerosols (Chapman et al. 2009). The estimation of the MAE (Table 1) shows a slight increase for the RF simulation in both episodes for maximum and minimum values of this statistical figure.

With respect to the correlation coefficients (Figs. 4 (e) and 4 (f) for the dust episode and 5 (e) and 5 (f) for the fires episode), both simulations show high levels (around 0.9) of this statistical figure during the dust episode, except for those areas with high levels of $AOD_{550}$, where the correlations are lower (even with negative correlations values close to -0.5). Conversely, for the fire episode, correlation values are close to 1 both for both cases (NRF and RF) over the entire domain, especially over the areas with high values of $AOD_{550}$.

When considering the improvement (or not) of the $AOD_{550}$ when including aerosol radiative feedbacks in the simulations, the difference in the MAE of the simulations between NRF and RF is estimated as defined in Eq. (2). For the dust episode (Fig 4 (b)), a slight improvement (worsening) is produced over the areas with medium (high/low) levels of $AOD_{550}$, taking these changes values between -0.09 and +0.12. For the fire episode (Fig 5 (b)), a worsening of MAE (difference NRF-RF of -0.02) is simulated close to the source of biomass burning aerosols. However, an improvement (up to +0.11) over areas further away from this source is estimated, which compensates the importance of including aerosol feedbacks in the simulations when assessing the improvement of worsening of simulations.

In the case of the $AE_{550/860}$ from MODIS, the results are analogous for both episodes. Low values (< 0.45, shown in Fig. 6 (a)) of this variable over the southeast of the domain are found. This, together with the high levels of $AOD_{550}$ (Fig. 4 (a)), is a clear indication that natural dust aerosols coming from the Saharan desert govern the $AOD_{550}$ levels here. On the other hand, for the fire episode (Fig. 7 (a)), the highest levels (around 1.6) are found over the north of Portugal, coincident with the fires areas, representing thus the emissions of biomass burning particles. Generally, for both simulations in both episodes, the model underestimates the high values of $AE_{550/860}$ and overestimates the low values.

For the dust episode, the MBE (Fig 6 (c) and (d)) minimum values are found of -0.65 and -0.62 for NRF and RF simulations (underestimation) and the maximum MBE takes values of 0.77 and 0.78, respectively (overestimation). Concerning the

correlation coefficient (Fig. 6 (e) and (f)), also for both simulations the value of this statistic is lower than for $AOD_{550}$. Over most of the domain negative values are found (around -0.7) and positive values found are low ($< 0.3$).

On the other hand, during the fires episode (Fig. 7.) MBE minimum values (underestimation) are found around -0.61 and -0.65 for NRF and RF simulation, respectively, and maximum MBE values around 0.68 and 0.66 for NRF and RF simulations.

With respect to the correlation coefficient, just for the dust episode, positive correlations ($> 0.5$) are located over the most of the domain, while negative correlations are estimated over the emission areas of biomass burning particles (with values around -0.8). However, for both episodes, a slight decrease for maximum and minimum MAE values (Table 1) are observed when the aerosol radiative feedbacks are taken into account.

At the same time, there is a slight improvement for RF simulations for the dust episode over the areas where the $AE_{550/860}$ is

overestimated (reaching values of improvement of MAE of 0.13) and a slight worsening (values of improvement of MAE around -0.09) over the areas where this variable is underestimated (Fig 5 (b)). For the fire episode, a slight improvement (values of improvement of MAE of 0.16) is found over the south-eastern part of the domain and a slight worsening (around -0.18) over the rest of the IP (Fig 6 (b)).

### 3.3 Model output vs. AERONET data

This section shows the results of the comparison between model output and AERONET data. First, a linear regression is estimated (Figs. 8, 9 and 10) and the correlation coefficients are calculated for the daily averages (Table 2). For AOD at different wavelengths during the dust episode, the results indicate that the stations where the model show higher skills are Barcelona and Sagres (maximum correlation coefficient 0.72) and, in for the fire episode, Caceres and Evora (maximum correlation coefficient 0.9 and 0.85, respectively). For $AE_{440/870}$, during the dust episode, the best-represented stations are

Caceres and Sagres (maximum correlation coefficient 0.62 and 0.57, respectively) and, for the fire episode, Autilla (0.75) and Evora (0.66). At this point, it is important to note that this comparison is obtained between a point (AERONET) and a cell (model outputs) covering the corresponding station coordinates following a nearest approach. In spite of the use of this approach, small errors on the spatial distribution of the model representation of the evaluated variables can appears, producing lower correlation coefficient values than the comparison with MODIS data, where the comparison is done cell (MODIS) versus

cell (model output) with approximately the same resolution. Results do not indicate a clear improvement or worsening for both variables in both episodes when including the aerosol radiative feedbacks in our modelling configuration.

High levels of AERONET AOD are found between 2-10 of July 2010 due to the dust outbreak in Barcelona and Sagres (Fig. 11 (a) & (c)). The time series of these stations have been selected as representative among all AERONET stations in the IP affected by the Saharan dust outbreak (see Supplementary Material for information on the rest of AERONET stations over the

IP). Maximum values of AERONET AOD occur between 7 and 10 July 2010. For $AOD_{1020}$, $AOD_{870}$ and $AOD_{675}$, the model underestimates the highest levels of AOD, represented by the minimum bias values (Table 2). On the other hand, between 2 and 6 of July 2010 (medium levels of AOD) the model overestimates the values of this parameter (Table 2). When an underestimation (overestimation) is produced, the bias is lower (higher) for lower wavelengths. Sagres station lacks of $AOD_{675}$

and $AOD_{440}$ data. Finally, the behaviour of $AOD_{440}$ in Barcelona is different from the other wavelengths due to the location of the station, close to a main street of the city where fine particles are emitted because of the road traffic.

For the fire episode, the shown stations are Caceres and Evora (see Supplementary Material for the rest of stations). In both stations, AOD presents the highest levels from 28 to 30 of July due to the wildfires occurred in Portugal (Table 2). Except for the first two days, the model tends to underestimate the AOD values. For all wavelengths the bias or error, in both stations, increases when the wavelength decreases.

Regarding the $AE_{440/870}$, for the dust episode the AERONET values show low values corresponding to large particles (generally between 0 and 1) in Sagres station, indicating the dust origin of the particles at this site. For the fires episode, values generally range between 1.5 and 2.5 at Evora station, revealing the small size of the biomass burning particles. Generally for all stations in both episodes, the model overestimates (underestimates) $AE_{440/870}$ values when there are low (high) values of this variable. Hence, the model strongly underpredicts the variability of this variable for the two configurations.

### 3.4. Model output vs. EARLINET data

Finally, the results of the comparison between model output and EARLINET data are shown. In this section, only the dust episode is studied because the only station with available data for both the study episodes in the IP during the year 2010 is Granada. At this site, dust has an important contribution to aerosol loads. Two specific days (6 and 12 July 2010) are shown for the sake of brevity, but this discussion is valid for other days of this episode. It is important to notice the differences between both discrete profile resolutions: the model profile with 33 levels from the ground to approx. 20 km; and the profile measurement, which much higher vertical resolution (7.5 m). So the results below should be considered mainly from a qualitative perspective. However, in order to provide a more quantitative approach, the MAE of the model versus lidar observations is estimated.

As for the particle backscatter (BSCAT) at 532nm for 6 July 2010 (Fig. 12(a)), the lidar detects a peak between 1.5 and $2 \times 10^{-6}$ $m^{-1}sr^{-1}$ around 3250 m above sea level caused by a dust layer. Although the model outputs overestimate the BSCAT values, simulations capture the profile of BSCAT. Although NRF and RF model configurations perform similarly, there is a slight improvement in the MAE of the vertical profile (estimated after Eq. 3) when the aerosol radiative feedbacks are taken into account (Fig. 12 (a)). Average MAE is $6.37 \times 10^{-7}$ and $6.22 \times 10^{-7}$ $m^{-1}sr^{-1}$ for NRF and RF simulations, respectively. Henceforth, the normalized MAE is improved by 2.4% when aerosol radiative feedbacks are included in WRF-Chem simulations.

For the BSCAT for 12 July 2010 (Fig. 12 (b)), the model overestimates the BSCAT values of the vertical profile, as aforementioned. However, the shape of the vertical profile is correctly reproduced. Mean MAE is $3.14 \times 10^{-7}$ and $3.12 \times 10^{-7}$ $m^{-1}sr^{-1}$ at 355nm for NRF and RF simulations, respectively; and $4.1 \times 10^{-7}$ $m^{-1}sr^{-1}$ at 532nm for both cases. Here, the improvement when including aerosol radiative feedbacks is very limited, and estimated as 0.63% and 0.14% at 355 and 532nm, respectively.

**4 Conclusions**

The use of on-line coupled models is one of the most powerful tools to understand the different processes influencing the climate system. In particular, for the study of atmospheric aerosol particles realistic simulation of the combined ARI and ACI are needed, irrespective of the aerosols source, where the interactions of aerosols, meteorology, radiation, and chemistry are coupled in a fully interactive manner. The use of modelling tools requires the observational study of physical, chemical and optical properties of aerosol particles to establish its behaviour and to assess how good these properties are represented in on-line coupled models.

In this study, two configurations including/not including (NRF/RF simulations) the aerosol radiative feedbacks have been assessed against a number of remote sensing observations for two episodes characterized by dust and biomass burning aerosols, respectively.

For the comparison between model output and MODIS data, the best-represented variable is AOD, with low values of mean bias and high values of correlation coefficient both for NRF and RF simulations. Discrepancies between simulations and observations can be ascribed to errors in the model estimation of the aerosol dry mass, the fraction of particles for a given mass or the water associated with aerosols. On the other hand, we should bear in mind the known errors from observations. The inclusion of the aerosol radiative feedbacks produces a slight improvement in the model representation for medium values of this variable and a worsening for the lowest and highest values. At the same time, the model output of AE representation leads to underestimate the variability of this variable. This occurred for both episodes and may be related to the fact that the size distribution of the aerosol function within WRF-Chem considers a medium size of particles, smaller for dust and larger for fire particles. The inclusion of aerosol feedbacks does not produce a clear benefit, taking into account the expensive computational cost required for including the ARI and ACI in the model. As well as for MODIS, for the comparison between model output and AERONET data, the results indicate that the best-represented variable is AOD. Generally, for both episodes, the model underestimates the levels of AOD, but the highest levels of this variable for dust episode are underestimated. It is important to note that the bias is usually higher for low wavelengths. In both episodes, the AE is overestimated for low levels and underestimated for high levels, since the modelled variability is strongly underestimated. For both variables, there is not a clear improvement of the model outputs for the aerosol radiative feedbacks simulation for any station in both episodes.

For the comparison between model output and EARLINET data, the results show a general slight improvement in the representation of vertical aerosol profiles when the aerosol radiative feedbacks are taken into account for all studied wavelength.

It is important to take into account these considerations to improve the time-efficiency when running the simulations, because the inclusion of aerosol radiative feedbacks in the simulations has a notable increase of the computational time. The improvements observed, in particular related to the vertical distribution of aerosols, justify the inclusion of aerosol radiative feedbacks in WRF-Chem on-line coupled model and the much higher time devoted to running the simulations.

**Acknowledgements**

The authors acknowledge the project REPAIR-CGL2014-59677-R of Ministerio de Economía y Competitividad and FEDER European program the support to carry out this research. The work has been possible thanks to the fellowship 19677/EE/14 funded by "Fundación Séneca-Agencia de Ciencia y Tecnología de la Región de Murcia", Programme "Jiménez de la Espada de Movilidad, Cooperación e Internacionalización", in the framework of II PCTIRM 2011-2014. Authors thank the support from EuMetChem COST ACTION ES1004 and the Air Quality Modelling Evaluation International Initiative (AQMEII).

This work also was supported by the University of Granada through the contract "Plan Propio. Programa 9. Convocatoria 2013", by the Andalusia Regional Government through project P12-RNM-2409, by the Spanish Ministry of Economy and Competitiveness through project CGL2013-45410-R and by the European Union's Horizon 2020 research and innovation programme through project ACTRIS-2 (grant agreement No 654109). The authors thankfully acknowledge the FEDER program for the instrumentation used in this work.

*Apendix. List of acronyms.*

| | |
|---|---|
| *ACI* | *Aerosol-cloud interactions* |
| *AE* | *Ångström Exponent* |
| *AERONET* | *AErosol Robotic NETwork* |
| *AOD* | *Aerosol Optical Depth* |
| *ARI* | *Aerosol-radiation interactions* |
| *BSCAT* | *Backscatter* |
| *DB* | *Deep Blue* |
| *DT* | *Dark Target* |
| *EARLINET* | *European Aerosol Research Lidar Network* |
| *ECMWF* | *European Centre for Medium-Range Weather Forecasts* |
| *EuMetChem* | *European framework for online integrated air quality and meteorology modelling* |
| *IFS-MOZART* | *Integrated Forecasting System - Model for ozone and related tracers* |
| *IP* | *Iberian Peninsula* |
| *IPCC* | *Intergovernmental Panel on Climate Change* |
| *IS4FIRES* | *Integrated monitoring and modelling system for wild-land fires* |
| *MACC-II* | *Monitoring Atmospheric Composition and Climate-Interim Implementation* |
| *MAE* | *Mean Absolute Error* |
| *MBE* | *Mean Bias Error* |
| *MEGAN* | *Model of Emissions of Gases and Aerosols from Nature* |
| *MODIS* | *Moderate Resolution Imaging Spectroratiometer* |
| *NRF* | *No radiative feedbacks* |
| *r* | *Correlation Coefficient* |
| *RF* | *Radiative feedbacks* |
| *RRTMG* | *Rapid Radiative Transfer Method for Global* |
| *S.L.* | *Significance Level* |
| *TNO* | *Netherlands Organization for Applied Scientific Research* |
| *YSU PBL* | *Yonsei University Planetary Boundary scheme* |
| *WRF-Chem* | *Weather Research and Forecasting model coupled with Chemistry* |

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

**Table 1: Maximum AERONET AOD at 1020, 870, 675 and 440 nm; maximum and minimum error or bias values for both episodes at different representative stations. Values in italics represents a better bias for the RF than for the NRF simulations.**

| | **Dust Episode** | | | | | | | | | | | | | | | |
|---|---|---|---|---|---|---|---|---|---|---|---|---|---|---|---|---|
| | **Barcelona Station** | | | | | | | | **Sagres Station** | | | | | | | |
| | *1020nm* | | *870nm* | | *675nm* | | *440nm* | | *1020nm* | | *870nm* | | *675nm* | | *440nm* | |
| Max AERONET AOD value | 0.43 | | 0.45 | | 0.47 | | 0.55 | | 0.39 | | 0.42 | | | | | |
| | NRF | RF | NRF | RF | NRF | RF | NRF | RF | NRF | RF | NRF | RF | NRF | RF | NRF | RF |
| Minimum bias value | -0.22 | -0.27 | -0.20 | -0.25 | -0.20 | -0.22 | -0.34 | -0.31 | -0.19 | -0.22 | -0.17 | -0.20 | | | | |
| Maximum bias value | 0.11 | *0.09* | 0.12 | *0.11* | 0.16 | *0.14* | 0.24 | *0.19* | 0.23 | *0.17* | 0.27 | *0.21* | | | | |
| | **Fire Episode** | | | | | | | | | | | | | | | |
| | **Caceres Station** | | | | | | | | **Evora Station** | | | | | | | |
| | *1020nm* | | *870nm* | | *675nm* | | *440nm* | | *1020nm* | | *870nm* | | *675nm* | | *440nm* | |
| Max AERONET AOD value | 0.23 | | 0.30 | | 0.38 | | 0.67 | | 0.35 | | 0.37 | | 0.41 | | 0.61 | |
| | NRF | RF | NRF | RF | NRF | RF | NRF | RF | NRF | RF | NRF | RF | NRF | RF | NRF | RF |
| Minimum bias value | -0.13 | *-0.12* | -0.17 | *-0.16* | -0.21 | *-0.19* | -0.40 | *-0.38* | -0.24 | -0.26 | -0.25 | -0.26 | -0.24 | -0.27 | -0.33 | -0.34 |
| Maximum bias value | 0.09 | 0.13 | 0.09 | 0.14 | 0.13 | 0.20 | 0.18 | 0.26 | 0.12 | 0.12 | 0.14 | 0.14 | 0.19 | 0.19 | 0.31 | *0.29* |

**Table 2: Values of correlation coefficient for the linear regression between AERONET and simulation daily means. Values in italic indicate the highest correlation coefficient among the different AOD/AE.**

| | | Autilla | | Barcelona | | Burjassot | | Caceres | | Evora | | Granada | | Huelva | | Malaga | | Sagres | |
|---|---|---|---|---|---|---|---|---|---|---|---|---|---|---|---|---|---|---|---|
| | | NRF | RF | NRF | RF | NRF | NRF | RF | NRF | RF | NRF | NRF | RF | NRF | RF | NRF | RF | NRF | RF |
| **Dust Episode** | $AOD_{1020}$ | 0.17 | 0.19 | 0.59 | 0.50 | *0.15* | 0.04 | 0.39 | 0.33 | *0.58* | 0.53 | 0.35 | 0.32 | 0.41 | 0.35 | 0.22 | 0.18 | 0.71 | 0.68 |
| | $AOD_{870}$ | 0.20 | 0.21 | 0.62 | 0.54 | 0.14 | 0.03 | *0.39* | 0.33 | 0.55 | 0.51 | 0.35 | 0.32 | 0.41 | 0.36 | 0.24 | 0.20 | *0.72* | 0.68 |
| | $AOD_{675}$ | 0.21 | *0.22* | 0.70 | 0.64 | 0.15 | 0.04 | 0.38 | 0.33 | 0.49 | 0.45 | 0.37 | 0.33 | 0.43 | 0.39 | 0.26 | 0.22 | | |
| | $AOD_{440}$ | 0.18 | 0.18 | 0.72 | *0.72* | 0.12 | 0.03 | 0.30 | 0.27 | 0.28 | 0.27 | *0.38* | 0.34 | *0.48* | 0.44 | *0.29* | 0.25 | | |
| | $AE_{440/870}$ | *0.09* | 0.08 | *0.16* | 0.04 | *0.15* | -0.05 | *0.62* | 0.51 | *0.38* | 0.36 | -0.23 | -0.32 | *0.22* | 0.08 | 0.20 | -0.26 | 0.35 | *0.57* |
| **Fire Episode** | $AOD_{1020}$ | 0.66 | 0.51 | 0.78 | 0.72 | 0.29 | 0.14 | 0.83 | 0.78 | 0.84 | 0.82 | 0.60 | 0.48 | 0.45 | 0.40 | 0.03 | 0.08 | 0.43 | 0.55 |
| | $AOD_{870}$ | 0.68 | 0.52 | *0.80* | 0.76 | 0.35 | 0.18 | 0.86 | 0.82 | *0.85* | 0.82 | 0.61 | 0.51 | 0.47 | 0.42 | 0.03 | 0.08 | 0.44 | *0.56* |
| | $AOD_{675}$ | 0.68 | 0.53 | 0.80 | 0.79 | 0.40 | 0.22 | 0.90 | 0.87 | 0.85 | 0.82 | *0.63* | 0.56 | 0.50 | 0.45 | 0.03 | *0.09* | | |
| | $AOD_{440}$ | *0.69* | 0.54 | 0.76 | 0.77 | *0.44* | 0.25 | *0.90* | 0.89 | 0.79 | 0.77 | 0.62 | 0.61 | *0.53* | 0.48 | 0.04 | 0.09 | | |
| | $AE_{440/870}$ | *0.75* | 0.71 | *0.50* | 0.46 | 0.06 | *0.18* | -0.03 | 0.05 | *0.66* | 0.56 | *0.27* | 0.16 | *0.25* | 0.20 | *0.3* | 0.24 | 0.28 | *0.28* |

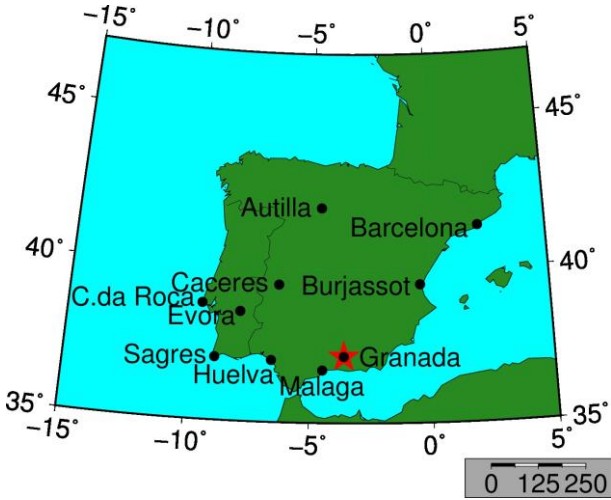

**Figure 1: Map of the distribution of the AERONET (points) and the EARLINET (star) stations.**

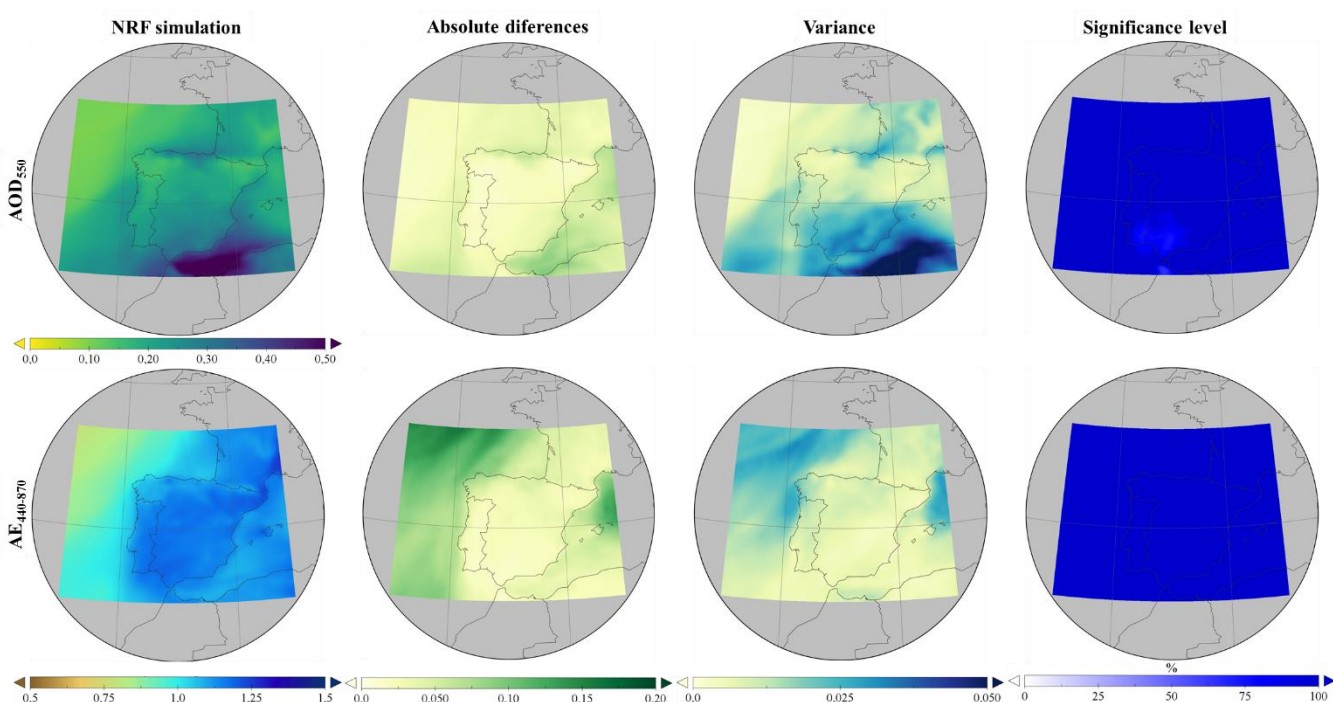

**Figure 2: Dust episode (temporal mean from 28 June to 12 July). Top: AOD550; bottom: AE440/870. From left to right: (a) modelled value of the variable, (b) value of the absolute differences between NRF-RF simulations, (c)variance value of NRF simulation and (d) Significance level (S.L.) values.**

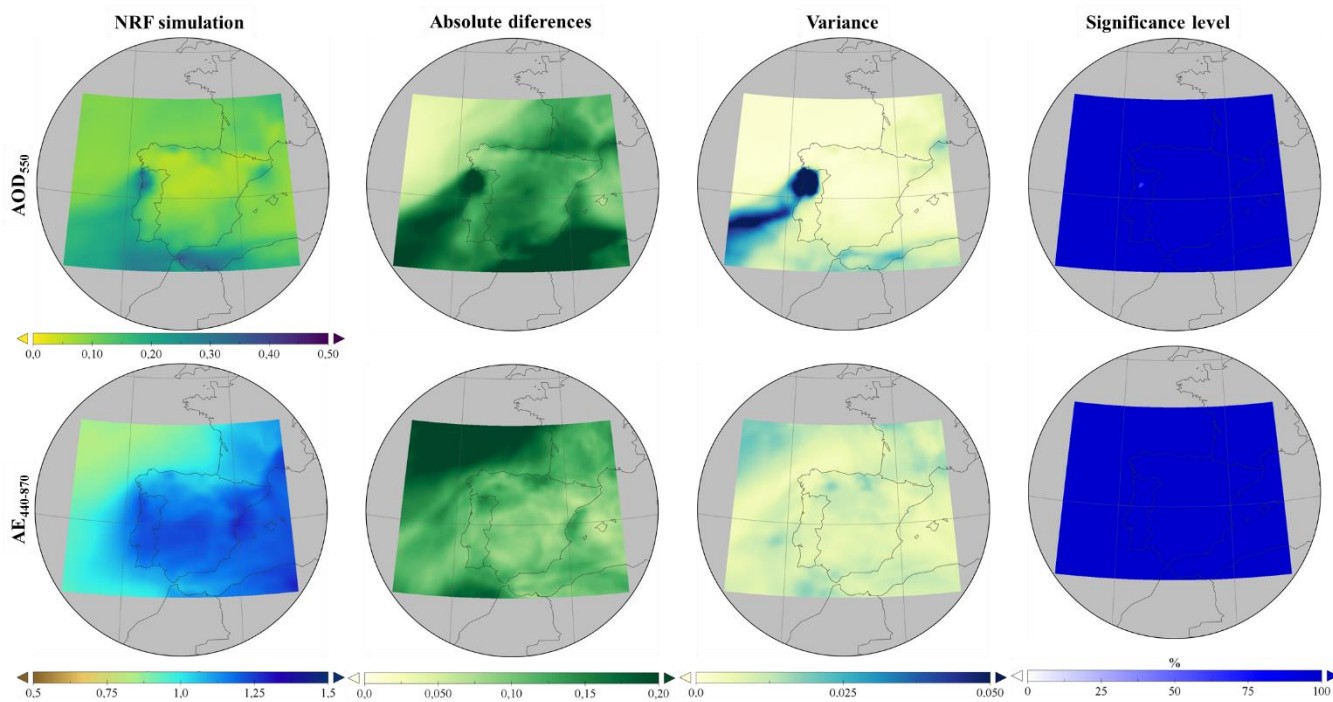

**Figure 3: Id. Fig. 2 for the fires episode, temporal mean from 25 July to 7 August.**

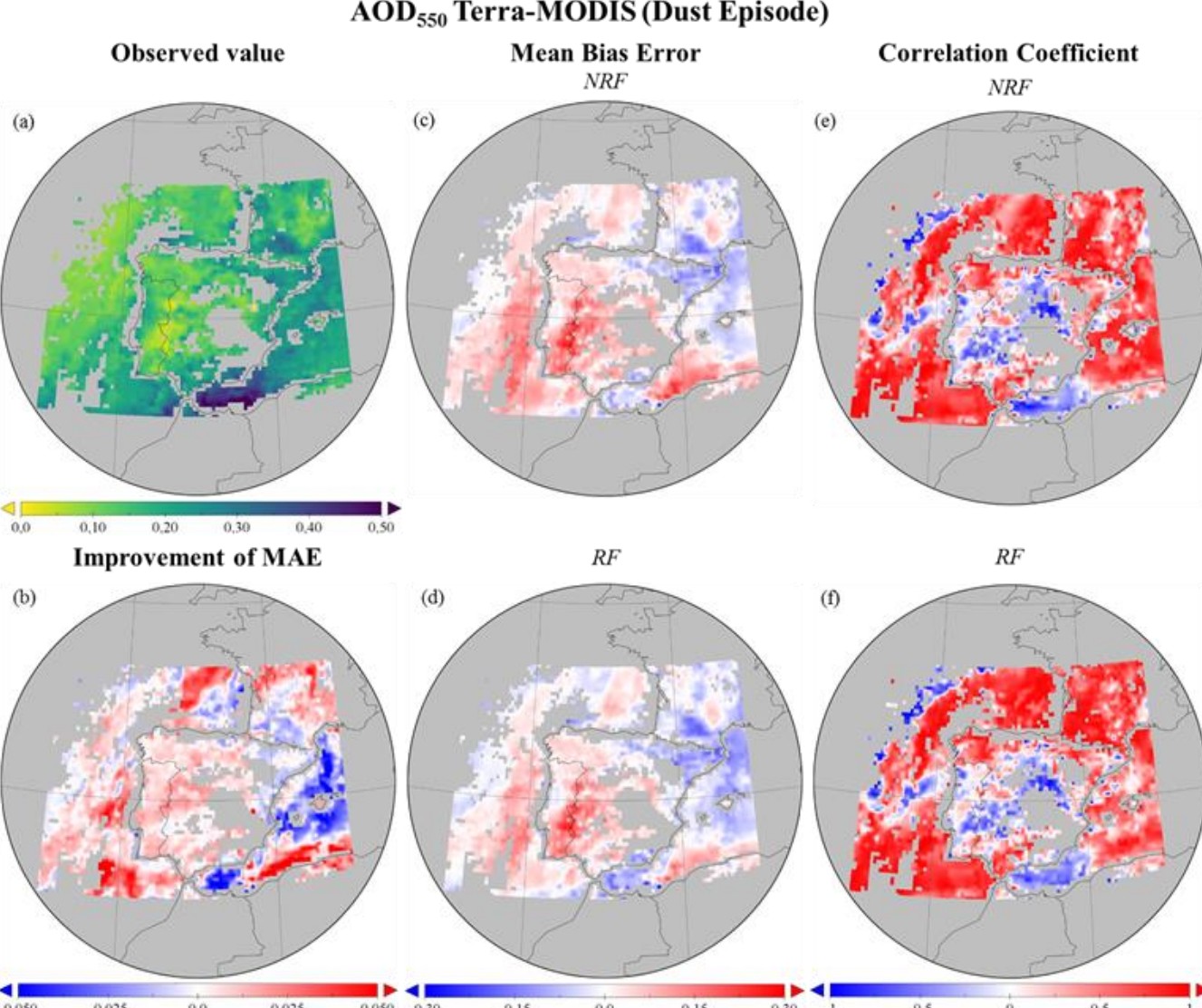

**Figure 4: Comparison of AOD$_{550}$ model output vs. AOD$_{550}$ from MODIS data for the dust episode (temporal mean from 28 June to 12 July). (a) AOD MODIS values. (b) Improvement of MAE due to the inclusion of RF (MAE in RF-NRF simulations). (c) and (d) MBE for NRF and RF simulations, respectively. (e) and (f) correlation coefficient for NRF and RF simulations.**

# AOD$_{550}$ Terra-MODIS (Fire Episode)

**Observed value**

**Mean Bias Error**
*NRF*

**Correlation Coefficient**
*NRF*

**Improvement of MAE**

*RF*

*RF*

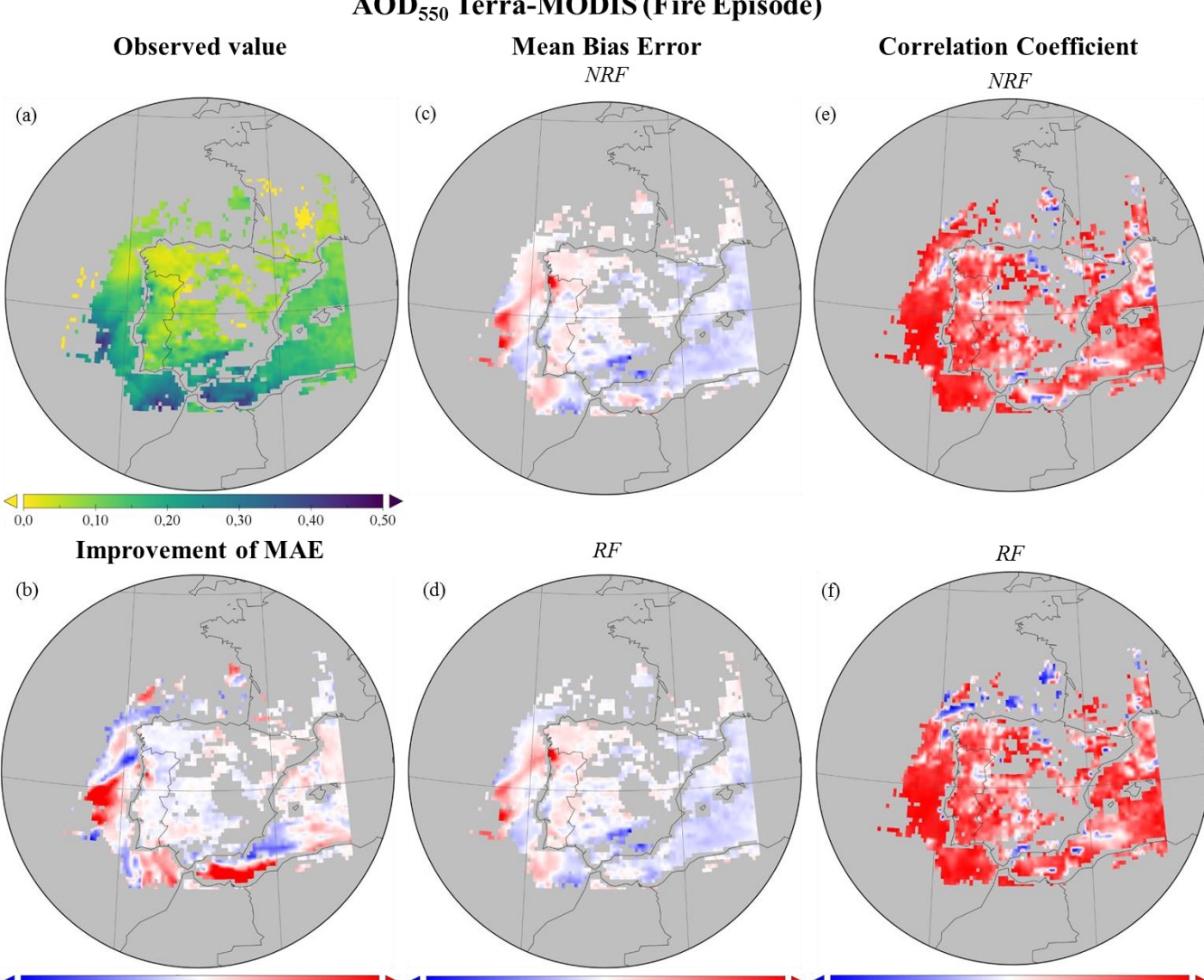

**Figure 5: As Fig. 4 for the fires episode (temporal mean from 25 July to 7 August).**

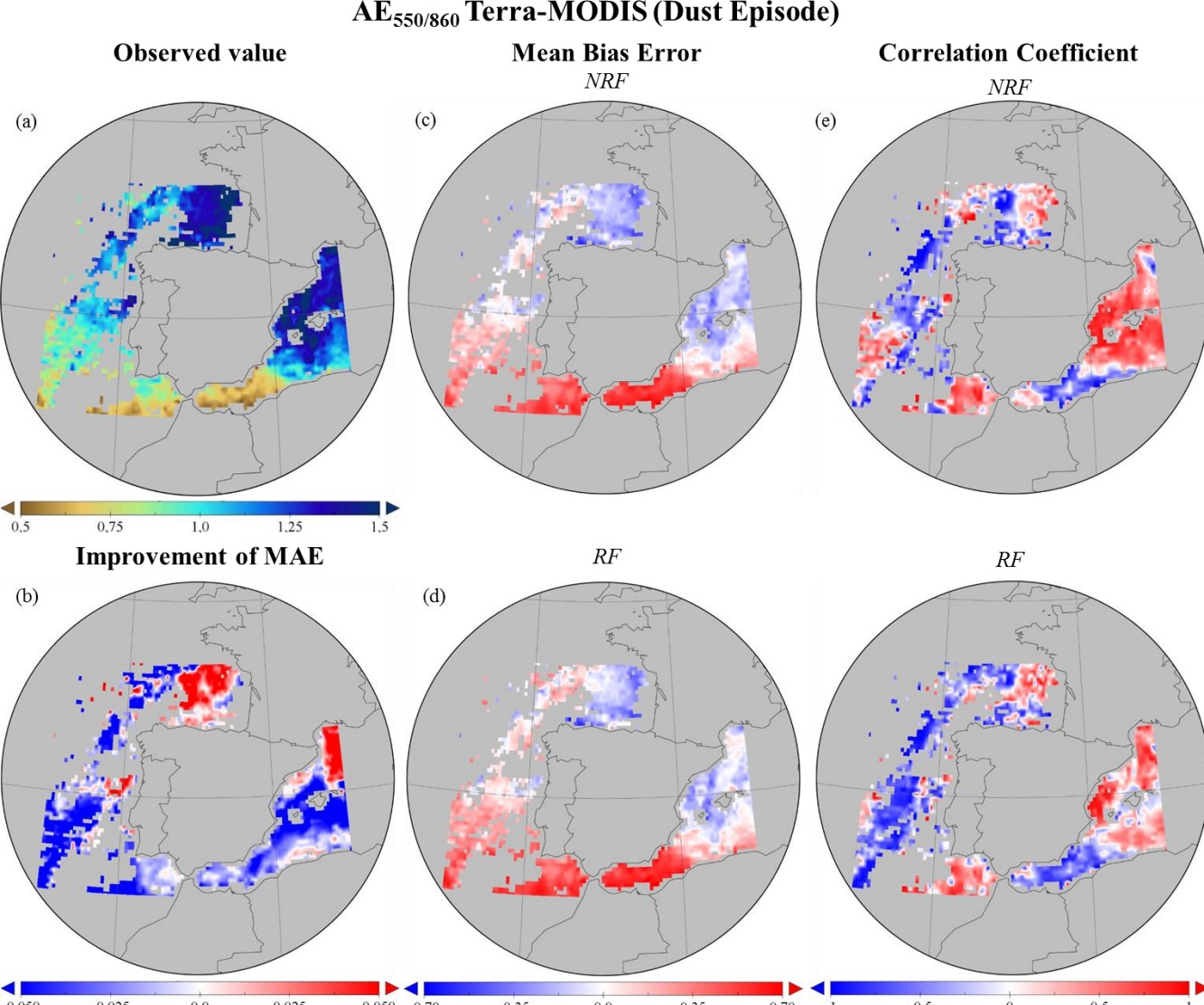

**Figure 6: Comparison of AE$_{550/860}$ model output vs. AE at same wavelength from MODIS data for the dust episode (temporal mean from 28 June to 12 July). (a) AE MODIS values. (b) Improvement of MAE due to the inclusion of RF (MAE in RF-NRF simulations). (c) and (d) MBE for NRF and RF simulations, respectively. (e) and (f) Correlation coefficient for NRF and RF simulations.**

# AE$_{550/860}$ Terra-MODIS (Fire Episode)

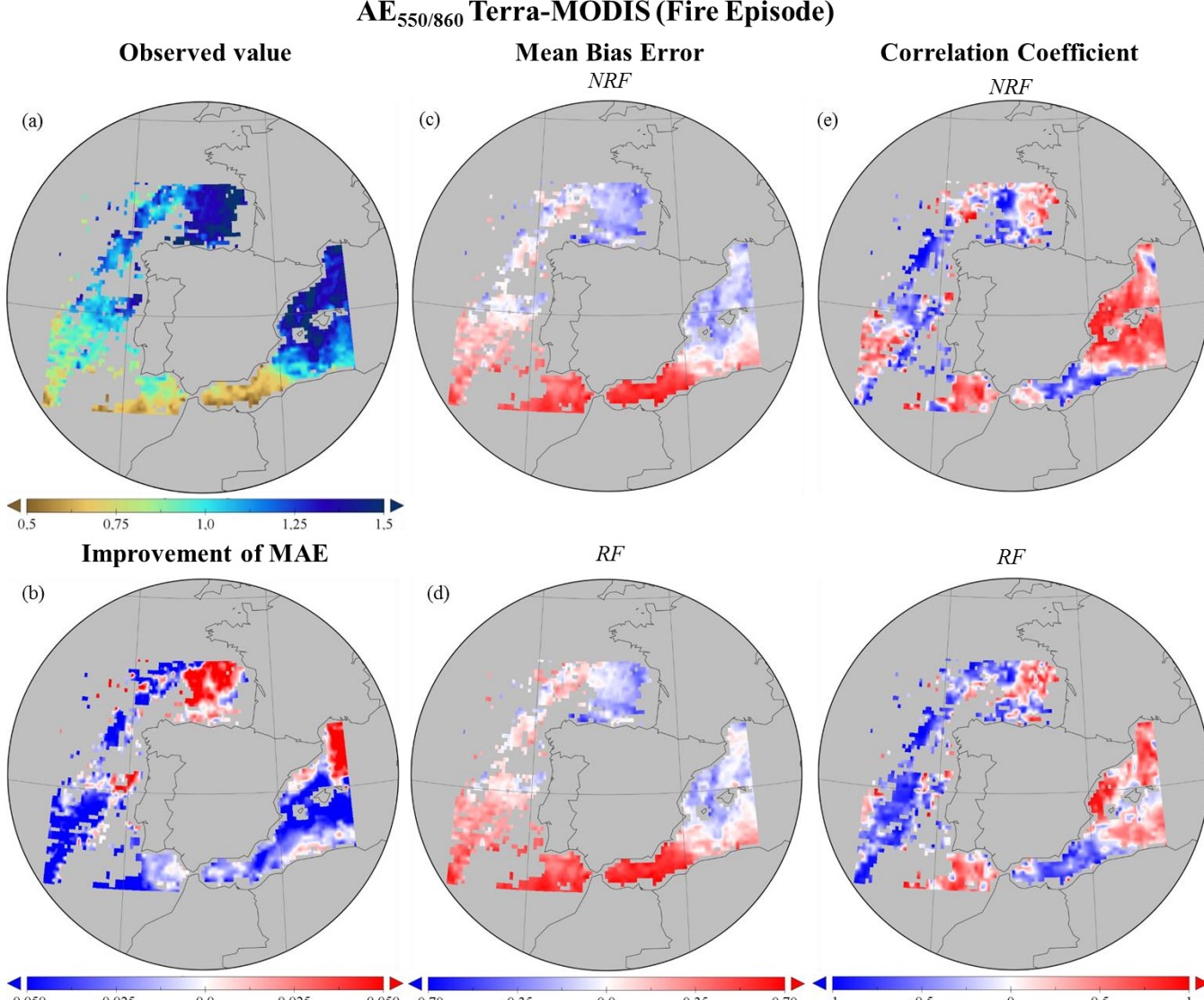

**Figure 7:** As Fig. 6 for the fires episode (temporal mean from 25 July to 7 August).

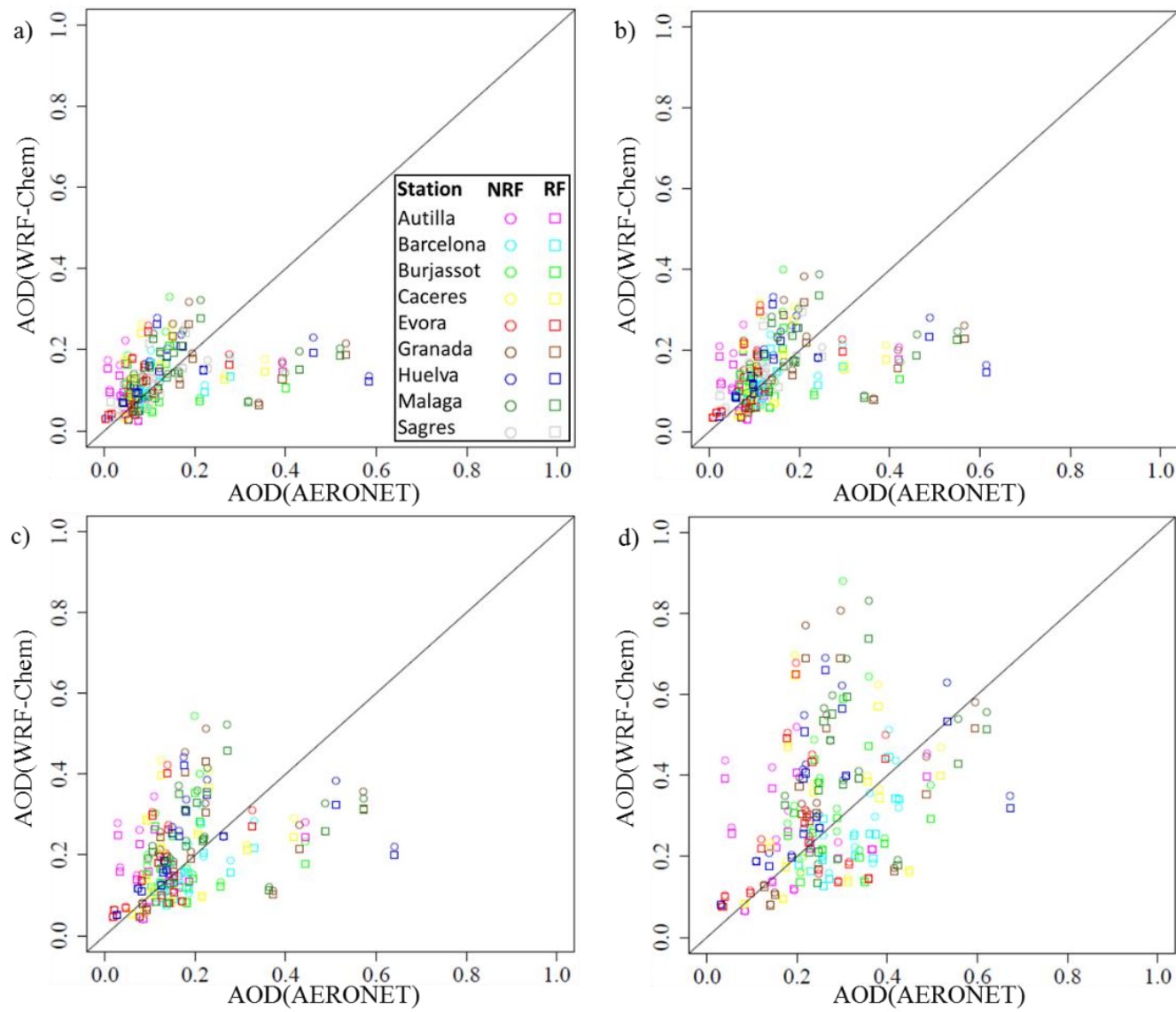

**Figure 8: Linear regression between AERONET (x) and simulations daily data (y; NRF in circles and RF in squares) for the dust episode (from 28 June to 12 July): (a) $AOD_{1020}$ (b) $AOD_{870}$ (c) $AOD_{675}$ and (d) $AOD_{440}$.**

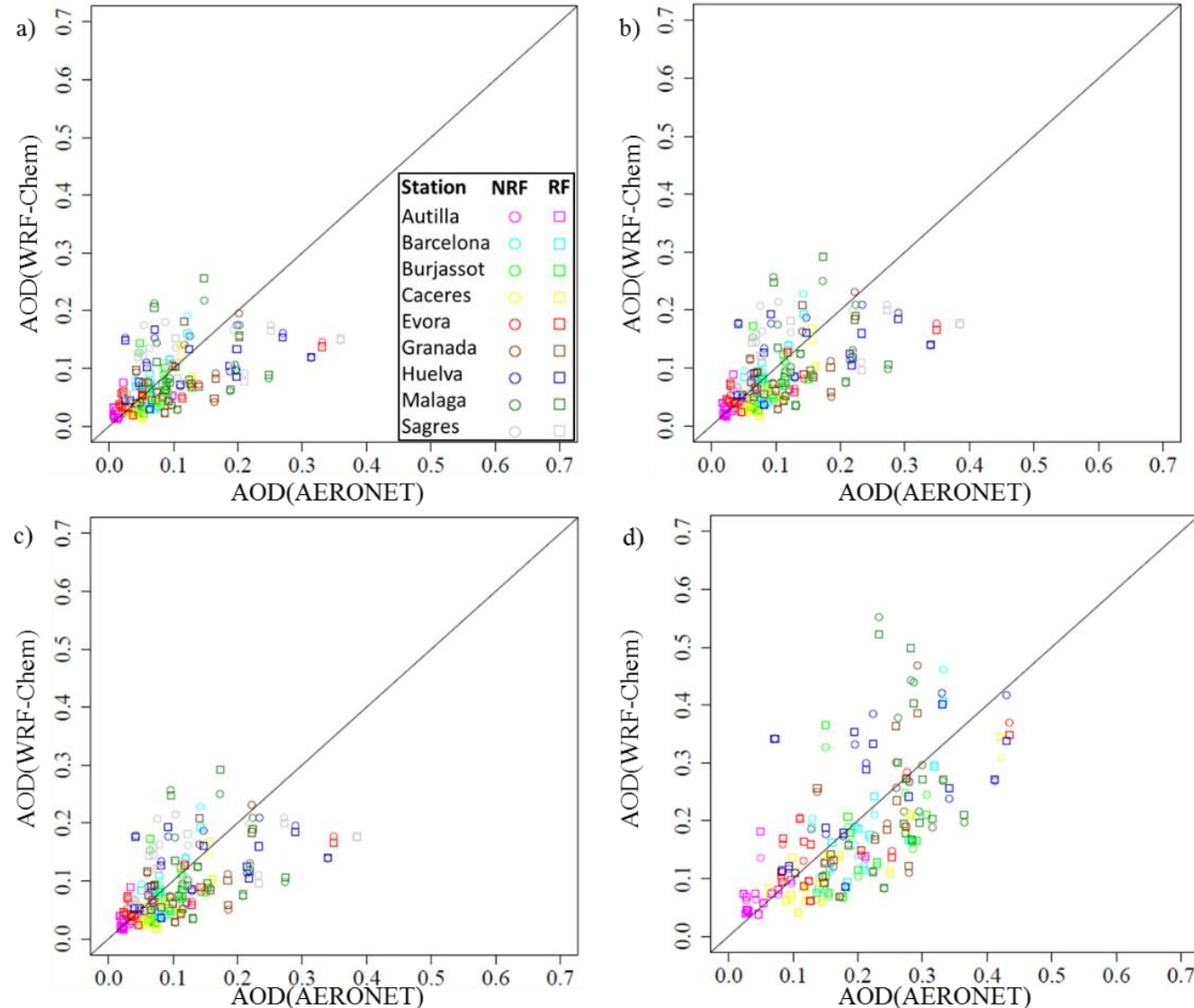

**Figure 9: As Fig. 8 for the fires episode (from 25 July to 7 August).**

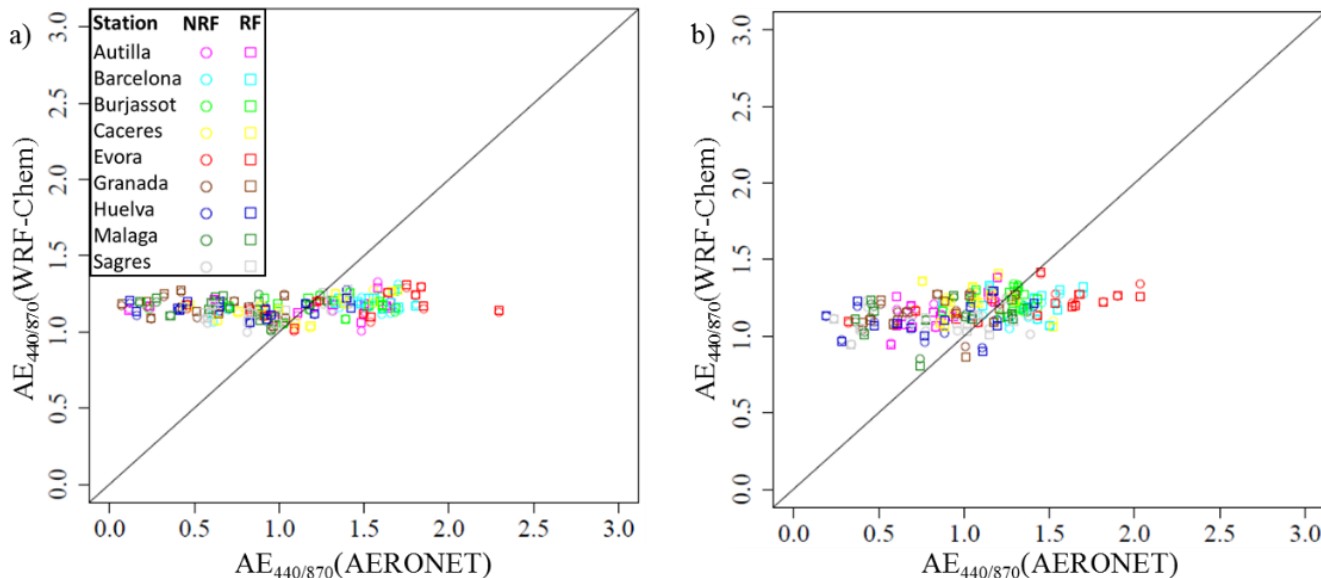

**Figure 10: Linear regression between AERONET (x) and simulations daily data (y, NRF in circles and RF in squares) of AE$_{440/870}$:**
**(a) dust episode (from 28 June to 12 July); (b) fire episode (from 25 July to 7 August).**

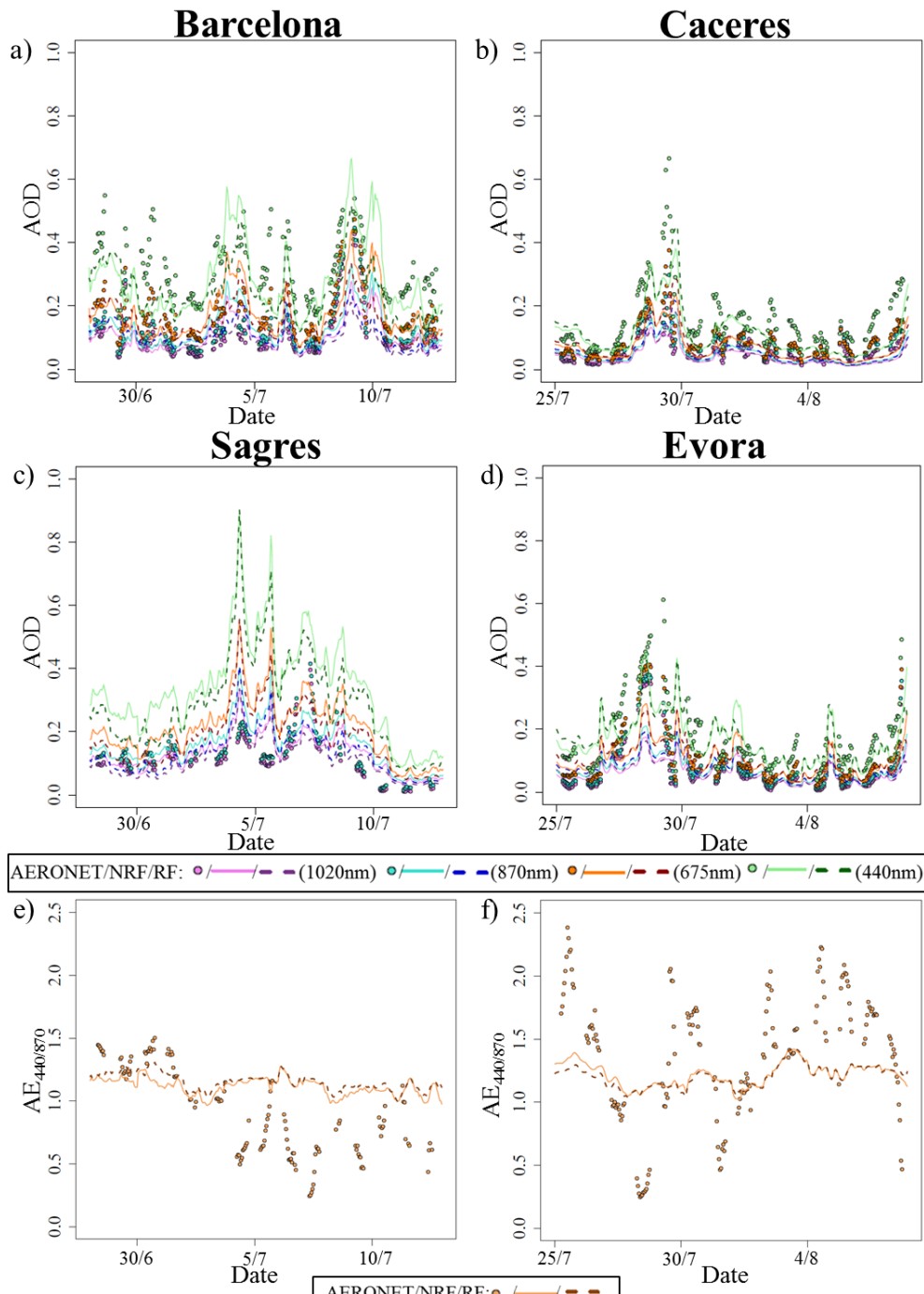

**Figure 11: AERONET (dots), NRF (line) and RF simulations (dashed line). AOD at different AERONET wavelengths: (a) and (c) Barcelona and Sagres stations for the dust episode (from 28 June to 12 July) and (b) and (d) Caceres and Evora stations for the fire episode (from 25 July to 7 August). (e) AE$_{440/870}$ in the Sagres station for the dust episode (from 28 June to 12 July) and for the (f) Caceres station for the fire episode (from 25 July to 7 August).**

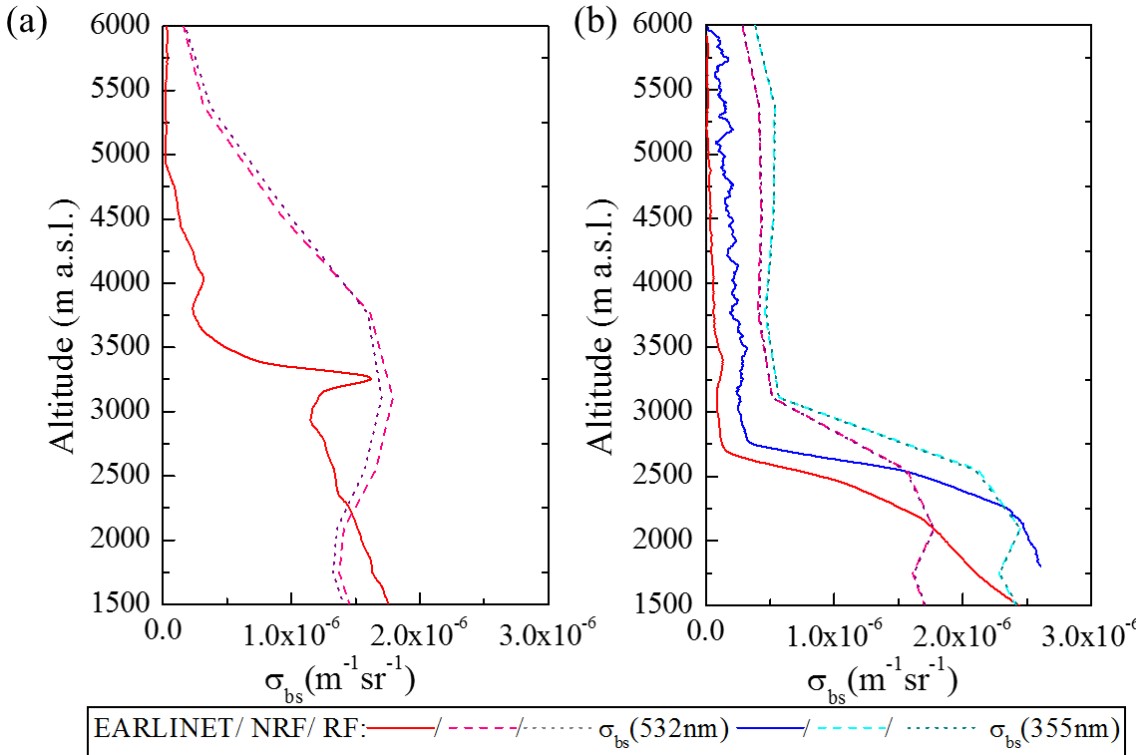

**Figure 12: EARLINET (line), NRF (dashed line) and RF simulations (dotted line) of the backscatter coefficient at 532nm and 355 nm. (a) For July 6, 2010 at 0200 UTC and (b) for July 12, 2010 at 1300 UTC.**