# Peer review of "Evaluating the representation of aerosol optical properties by an online coupled model over the Iberian Peninsula"

_Atmospheric Chemistry and Physics, 2016_

## Referee Comment (RC1) · Anonymous Referee #1 · 4 Aug 2016

General comments: From my understanding, the paper "Assessment of the radiative effects of aerosol in an on-line coupled model over the Iberian Peninsula" presents results of a numerical study focused on the sensitivity of atmospheric aerosol particles optical properties over Iberian Peninsula (IP), nominally Aerosol Optical Depth (AOD), Ångström Exponent (AE), and backscattering vertical profile, to the feedbacks induced by the aerosol direct and indirect radiative effects. Two numerical experiments using the WRF-Chem model were performed, one simulating AOD and AE fields and aerosol vertical distribution turning off aerosol particles radiative effects in the model, and a second experiment turning on the aerosols radiative effects, both the direct and the indirect. To assess the impacts (improvement or deterioration) of the on-line coupling

of aerosol direct and indirect radiative effects on the simulation of AOD and AE field over IP, observational data of AOD and AE were taken from database products of the MODIS aboard of Terra and Aqua orbital platforms and Aerosol RObotic NETwork (AERONET) ground based stations in the IP. As observational reference for aerosol vertical distribution they have used data from the EARLINET station in Granada. The study is focused on two distinct aerosol scenarios over IP, one consists of an episode of Saharan dust transport toward the southern region of IP, and the second of a biomass burning event that occurred in the north region of Portugal. Although the scientific goal of the paper is of significant relevance, the paper exposure needs improvements. Beginning from the title, in my view, the authors did not assess the aerosols radiative effects as the title suggest. So far, their major focus has been on the response of aerosol optical properties field over IP, mainly AOD and EA, to the on-line coupling of the aerosol radiative effects in the model. Therefore, I think there is a need to adequate the title in order to accurately express the paper goal and content. For the sake of clarity, the authors should make clear the distinction between what they refer as radiative feedbacks and the aerosol radiative effects. Given the role of the numerical experiments on the manuscript goals and conclusions, there is not much discussion on the model system and simulations configurations, physical and chemical modules, leaving it to references. Further details are needed, especially as regard to the model aerosol microphysical and optical modules, which have relevant impact on the variables analysed, and indirect and direct radiative effects parametrization. Also, there is not much discussion about the mechanisms that drive the feedbacks induced by the on-line simulation of the aerosol direct or indirect radiative effects on the aerosol field over IP. Results and discussions are essentially describing the discrepancies between simulations results and observation without further discussion on the potential drivers.

Specific comments: Page 2, Line 09: "Light-absorbing aerosols such as biomass burning exert a warming influence. . ." That may be true for black carbon aerosol particles, however biomass burning aerosol plumes are not only composed by black carbon. Biomass burning plume as a whole may have a cooling effects (references example:

[Figure]

Schafer et al., 2002, Observed reductions of total solar irradiance by biomass-burning aerosols in the Brazilian Amazon and Zambian Savanna, GRL, Volume: 29 Issue: 17; Procopio et al., 2004, Multiyear analysis of amazonian biomass burning smoke radiative forcing of climate, GRL, Volume: 31 Issue: 3). Page 3, Line 14: Please, include specifically which modelling output are you refering to.

Page 5: Emissions sources are discussed here, however nothing is said about the dust emission, one of the aerosol type focus of the study.

Page 5, Line 17: "... aerosol particles are represented by two lognormal size distributions, corresponding to an Aitken mode and an accumulation mode...": Considering that an event of Saharan dust outbreak is analysed, a coarse mode consideration wouldn't be relevant? The absence of a coarse mode aerosol in the model parametrization certainly helps to explain the discussed model difficulty to simulate Angstrom Exponent variability.

Page 6, Line 18: MODIS Angstrom Exponent is only available for ocean region? If yes, so the analysis was not restricted to Iberian Peninsula, but also over the surround sea and ocean.

Page 8, Line 1- 5: Certainly MODIS retrievals have issues, but also it would be important to discuss the modelling issues that can contribute to the discrepancies.

Page 8, Line 1 – 2: Is the correlation coefficient obtained from model simulation comparison with MODIS data distinct from that calculated for the comparison between model simulation against AERONET? If so, why is correlation coefficients for model x MODIS much higher than correlation coefficients for model x AERONET (Table 3)? How does MODIS AOD compare with AERONET stations AOD?

Page 10, Line 4 – 5: The inclusion of more days in the analysis may provide a better analysis from the statistical perspective. Page 18, Table 1: A map of the distribution of the AERONET sites and the EARLINET station in Granada would be helpful to the

readers to follow the discussions. For example, that can be done in one of the AOD field map from the simulation.

Technical corrections: Although the comprehension of the manuscript is not affected, I would recommend that the authors make use of an editing service, so that the writing can be improved. There are many sentences that need improvements; here I list some of them.

Page 1, Abstract first line: "...over the Earth's climate..." to "... on the Earth's climate..."

Page 2, Line 3: "...cause changes are: (1) scattering and absorption of solar radiation..." to "...cause changes are: (1) scattering and absorbing solar and terrestrial radiation..." Dust aerosol in particular may affect terrestrial radiation.

Page 2, Line 14: The sentence "The large uncertainty quantifying these ..." read better as "The uncertainty quantification of these aerosol effects on the Earth radiative budget is much higher..."

Page 3, Line 22: "...altering the global budget indirectly..." to "...altering the global energy budget indirectly..."

Page 3, Line 27: "The grid size is 6000 cells..." to "The grid size consists of 6000 cells..."

Page 7, Line 18: "We can then state then that the changes..." to "We can then state that the changes..." Page 9, Line 16: replace "...10 (a) & (c)..." to "...10 (a) and (c)..."

Page 9, Line 21: " Sagres stations ..." to "Sagres station..."

Page 10, Line 3: "Several specific days ..." to "Two specific day..."

Page 19, Table 1 and 2: Part of the table at the right side is missing.

Recommendation for the figures legends: Include the period over which mean field

[Figure]

AOD and AE are calculated and avoid abbreviations such as S.L (significant level)

---

## Referee Comment (RC2) · Anonymous Referee #2 · 22 Oct 2016

The objective of this paper is to quantify the aerosol radiative feedback for the Iberian Peninsula for some pollution episodes. A coupled meteorology-chemistry model WRF-Chem was used to simulate gas/aerosol chemistry, aerosol-radiation-cloud interactions for two case studies. These cases were focused on a Saharan dust break and a wildfire episodes. The model results were evaluated using various remote sensing datasets. The subject of this study is relevant for publication in ACP. It is crucial to accurately estimate feedback of aerosols from different sources to radiation budget over the region.

In my opinion authors need to improve the quality of presentation of the modeling framework, the model results and evaluations in the paper before it is accepted to ACP.

The section 2.1 provides limited information about the WRF-Chem model setup used

in the study. Which gas chemistry, microphysics etc. options were used in the model?

Why did the authors choose the SORGAM module? It's well known that the SORGAM drastically underestimates secondary organic aerosol (SOA) concentrations, consequently total aerosol concentrations. There are versions of the MADE aerosol scheme coupled to new SOA schemes in WRF-Chem (e.g. Tuccella et al., 2015).

The authors need to provide more details on how the aerosol-radiation and aerosol-cloud interactions are parameterized in their version of WRF-Chem. These details could help to better interpret the model-observation discrepancies.

The model was run on 23km resolution. This is a relatively coarse model grid. It doesn't allow simulating land-sea breeze and other mesoscale circulations. Moreover, in such resolution there are more parameterized (by cumulus parameterization) clouds in the model. Since the model doesn't treat aerosol-cloud feedback in cumulus parameterization, the overall ACI effect can't be captured by these model settings.

Another uncertainty stems from using ECMWF analysis fields for the meteorological initial and boundary conditions in the regional WRF-Chem modeling. The ECMWF model assimilates met. observations, which might be already affected by those dust and fire aerosols. Hence, the base WRF-Chem model case implicitly may already include some of the aerosol feedback. I understand that it's hard to set up a "perfect" regional modeling framework to study the aerosol-meteorology interactions, however this issue needs to be mentioned in the paper.

I don't see much discussions of the simulated ACI effect in the paper. For clarity it'd better to show three model cases - w/o any aerosol feedback, with aerosol feedback on radiation and with aerosol feedback on radiation+clouds, and discuss them more thoroughly.

Another missing piece in this paper is lack of evaluations of the simulated aerosol concentrations. Thus, it's hard to interpret AOD comparisons given the lack of information

about the model's skill to simulate aerosol mass concentrations in dust and smoke plumes.

Minor comments: Authors use many abbreviations in the text. I suggest adding a table showing all of them in one place.

page 9: correct "values shows"

page 11: correct "fires particles"

References section: The paper by Iacono et al. is entered twice.

REFERENCES: Tuccella, P., G. Curci, G. A. Grell, G. Visconti, S. Crumeyrolle, A. Schwarzenboeck and A. A. Mensah (2015). "A new chemistry option in WRF-Chem v. 3.4 for the simulation of direct and indirect aerosol effects using VBS: evaluation against IMPACT-EUCAARI data." Geoscientific Model Development 8(9): 2749-2776.

---

## Author Comment (AC1) · 22 Nov 2016

Rev. #1: Beginning from the title, in my view, the authors did not assess the aerosols radiative effects as the title suggest. So far, their major focus has been on the response of aerosol optical properties field over IP, mainly AOD and EA, to the on-line coupling of the aerosol radiative effects in the model. Therefore, I think there is a need to adequate the title in order to accurately express the paper goal and content.

*A: As suggested by the Rev. #1, the title has been changed by "Evaluating the representation of aerosol optical properties by an on-line coupled model over the Iberian Peninsula".*

Rev. #1: For the sake of clarity, the authors should make clear the distinction between what they refer as radiative feedbacks and the aerosol radiative effects.

*A: An effort has been made in the abstract and the introduction to define radiative effects and radiative feedbacks. Aerosol radiative effects refer to direct and semi-direct effects, produced by the aerosol-radiation interactions (ARI); and indirect effects, produced by aerosol-cloud interactions (ACI) (as described in the submitted manuscript). These radiative effects produce feedbacks to meteorology/emissions, which are called aerosol radiative feedbacks.*

Rev. #1: Given the role of the numerical experiments on the manuscript goals and conclusions, there is not much discussion on the model system and simulations configurations, physical and chemical modules, leaving it to references. Further details are needed, especially as regard to the model aerosol microphysical and optical modules, which have relevant impact on the variables analysed, and indirect and direct radiative effects parametrization. Also, there is not much discussion about the mechanisms that drive the feedbacks induced by the online simulation of the aerosol direct or indirect radiative effects on the aerosol field over IP.

*A detailed description of the aerosol, microphysical and optical modules as well as the previous description of the representation of aerosol-radiation-clouds interactions has been included in the revised version of the manuscript:*

[revised manuscript text omitted]

Rev. #1: Results and discussions are essentially describing the discrepancies between simulations results and observation without further discussion on the potential drivers.

*A: For AOD, discrepancies between simulations and observations can be ascribed to errors in the model estimation of the aerosol dry mass, the fraction of particles for a given mass or the water associated with aerosols. On the other hand, the known errors from observations have to be considered (as indicated in the manuscript, page 8: "we should bear in mind that this fact may be conditioned by the MODIS underestimation of $AOD_{550}$ levels for high loads of this type of particles, which has been reported by Chu et al., 2002; Levy et al., 2005 and Remer et al., 2005, among others"). For AE, discrepancies can be ascribed to an underestimation on the variability of particles size.*

*The aforementioned comment has been incorporated into the Conclusions section (pages 11-12).*

**Specific comments:**

Rev. #1: Page 2, Line 09: "Light-absorbing aerosols such as biomass burning exert a warming influence. . ." That may be true for black carbon aerosol particles, however biomass burning aerosol plumes are not only composed by black carbon. Biomass burning plume as a whole may have a cooling effects (references example: Schafer et al., 2002, Observed reductions of total solar irradiance by biomass-burning aerosols in the Brazilian Amazon and Zambian Savanna, GRL, Volume: 29 Issue: 17; Procopio et al., 2004, Multiyear analysis of amazonian biomass burning smoke radiative forcing of climate, GRL, Volume: 31 Issue: 3).

*A: The sentence has been rephrased in order to take into account the reviewer comment. E.g. "Light-absorbing aerosols such as black carbon, which are a component of biomass burning, exert a warming influence (e.g. Jacobson, 2001)."*

Rev. #1: Page 3, Line 14: Please, include specifically which modelling output are you refering to.

*A: Modelling outputs are aerosol optical properties (aerosol optical depth, AOD and Angstrom exponent, AE). This has been clarified in the revised manuscript.*

Rev. #1: Page 5: Emissions sources are discussed here, however nothing is said about the dust emission, one of the aerosol type focus of the study.

*A: WRF-Chem predicts online dust emission as a function of the land usage information and the simulated meteorological fields. In this work and following Shaw et al. (2008), dust emission flux (G) depends on: an empirical proportionality constant estimated based on regional specific data (C); the vegetation mask accounting for vegetation type (α); the friction velocity ($u_*$); the threshold friction velocity below which dust emission does not occur ($u_{*t} = 20\ cm\ s^{-1}$ following Shaw et al., 2008); and the soil wetness factor accounting for soil moisture ($f_w$).*

$$G = \alpha C u_*^4 \left(1 - \frac{f_w u_{*t}}{u_*}\right)$$

*This has been clarified in the revised manuscript.*

Shaw, W., Allwine, K. J., Fritz, B. G., Rutz, F. C., Rishel, J. P., and Chapman, E. G.: An evaluation of the wind erosion module in DUSTRAN, Atmos. Environ., 42, 1907–1921, 2008.

Rev. #1: Page 5, Line 17: ". . . aerosol particles are represented by two lognormal size distributions, corresponding to an Aitken mode and an accumulation mode. . .": Considering that an event of Saharan dust outbreak is analysed, a coarse mode consideration wouldn't be relevant? The absence of a coarse mode aerosol in the model parametrization certainly helps to explain the discussed model difficulty to simulate Angstrom Exponent variability.

*A: The model includes coarse aerosols in the model parameterization. This point has been clarified in the revised version of the manuscript. E.g. "Here aerosol particles are represented by three log-normal size distributions, corresponding to an Aitken mode (nucleation mode 0.1 µm diameter), and accumulation mode (0.1 – 2 µm), and a coarse mode (> 2 µm) (Forkel et al. 2012)."*

Forkel, R., Werhahn, J., Hansen, A. B., McKeen, S., Peckham, S., Grell, G., & Suppan, P. (2012). Effect of aerosol-radiation feedback on regional air quality–a case study with WRF/Chem. Atmospheric environment, 53, 202-211.

Rev. #1: Page 6, Line 18: MODIS Angstrom Exponent is only available for ocean region? If yes, so the analysis was not restricted to Iberian Peninsula, but also over the surround sea and ocean.

*A: As the reviewer indicates this area covers the Iberian Peninsula and the surrounding sea and ocean. For this reason, a better description of the study area has been done in the revised version of the manuscript.*

Rev. #1: Page 8, Line 1- 5: Certainly MODIS retrievals have issues, but also it would be important to discuss the modelling issues that can contribute to the discrepancies.

*A: Generally a too high predicted AOD by the model can be explained by either too much aerosol dry mass present in the model, too large fraction of small particles for a given mass,*

*or due to an excess of water associated with the aerosols (Chapman et al. 2009). This has been clarified in the revised manuscript (Page 8).*

*Chapman, E. G., Gustafson Jr., W. I., Easter, R. C., Barnard, J. C., Ghan, S. J., Pekour, M. S., and Fast, J. D.: Coupling aerosol-cloud-radiative processes in the WRF-Chem model: Investigating the radiative impact of elevated point sources, Atmos. Chem. Phys., 9, 945-964, doi:10.5194/acp-9-945-2009, 2009.*

Rev. #1: Page 8, Line 1 – 2: Is the correlation coefficient obtained from model simulation comparison with MODIS data distinct from that calculated for the comparison between model simulation against AERONET? If so, why is correlation coefficients for model x MODIS much higher than correlation coefficients for model x AERONET (Table 3)? How does MODIS AOD compare with AERONET stations AOD?

*A: Correlation coefficients for model x AERONET are obtained from a comparison between a point (AERONET) and a cell (model outputs) covering the corresponding station coordinates following a nearest neighbour approach. In spite of the use of this approach, small errors on the spatial distribution of the model representation of the evaluated variables can appear, producing lower correlation coefficient values than the comparison with MODIS data, where the comparison is done cell (MODIS) vs. cell (model) with approximately the same resolutions. This comment has been introduced in the paper (Page 10)*

*Regarding the second question (MODIS AOD vs. AERONET AOD), the comparison between MODIS AOD and AERONET stations AOD has been done by using the revised protocol developed by Petrenko et al. (2012), where satellite and sun photometer are compared within a spatial radius of ±25 km and a temporal interval of ±30 min. A valid collocation is one where there are at least three MODIS pixels and two sun photometer measurements within the spatial/temporal window. For Collection 6 (C6), the correlation is R = 0.86, and that 69.4 % of MODIS AOD fall within expected uncertainty of ± (0.05 + 15 %) (Levy et al. 2013).*

[Figure]

*Figure 1. Left: frequency scatter plots for AOD at 0.55 µm over dark-land compared to AERONET, plotted from 6 months of Aqua (January and July; 2003, 2008 and 2010), computed with C6 algorithm (b). One-one lines and EE envelopes ± (0.05 + 15 %) are plotted as solid and dashed lines. Collocation statistics are presented in each panel. Right: the same information plotted as AOD error (MODIS-AERONET) versus AERONET, broken into equal number bins of AERONET AOD (d). One-one line (zero error) is dashed and EE envelopes are solid. For each box-whisker, its properties and what they represent include: width is 1-σ of the AOD bin, whereas height, whiskers, middle line and red dots are the 1-σ, 2-σ, mean and median of the AOD error, respectively*

*Petrenko, M., Ichoku, C., and Leptoukh, G.: Multi-sensor Aerosol Products Sampling System (MAPSS), Atmos. Meas. Tech., 5, 913–926, doi:10.5194/amt-5-913-2012, 2012.*

*Levy, R. C., Mattoo, S., Munchak, L. A., Remer, L. A., Sayer, A. M., Patadia, F., and Hsu, N. C.: The Collection 6 MODIS aerosol products over land and ocean, Atmos. Meas. Tech., 6, 2989-3034, doi:10.5194/amt-6-2989-2013, 2013.*

Rev. #1: Page 10, Line 4 – 5: The inclusion of more days in the analysis may provide a better analysis from the statistical perspective.

*A: The reviewer is right in his/her appreciation; however, bearing in mind the high computational costs and the framework of this work (related to EuMetChem Cost Action ES1004), just two episodes for the year 2010 have been included. In EuMetChem, the objective is to evaluate two important aerosol episodes differing in the type of aerosol (biomass burning vs. dust). The inclusion of more days in the analysis would imply a high computational cost and would not represent such extreme events as the ones considered in this work.*

Rev. #1: Page 18, Table 1: A map of the distribution of the AERONET sites and the EARLINET station in Granada would be helpful to the readers to follow the discussions. For example, that can be done in one of the AOD field map from the simulation.

*A: Table 1 with stations coordinates has been changed by a map of the distribution of AERONET and EARLINET stations in the revised version of the manuscript.*

**Technical corrections:**

Rev. #1: Although the comprehension of the manuscript is not affected, I would recommend that the authors make use of an editing service, so that the writing can be improved. There are many sentences that need improvements; here I list some of them:

*A: Please find below the list of recommendations of the Reviewer and the corrections made.*

Page 1, Abstract first line: ". . .over the Earth's climate..." to ". . . on the Earth's climate. . ." *(Done)*
Page 2, Line 3: ". . .cause changes are: (1) scattering and absorption of solar radiation. . ." to ". . .cause changes are: (1) scattering and absorbing solar and terrestrial radiation. . ." Dust aerosol in particular may affect terrestrial radiation. *(Done)*
Page 2, Line 14: The sentence "The large uncertainty quantifying these . . ." read better as "The uncertainty quantification of these aerosol effects on the Earth radiative budget is much higher. . ." *(Done)*
Page 3, Line 22: ". . .altering the global budget indirectly. . ." to ". . .altering the global energy budget indirectly. . ." *(Done)*
Page 3, Line 27: "The grid size is 6000 cells. . ." to "The grid size consists of 6000 cells. . ." *(Done)*
Page 7, Line 18: "We can then state then that the changes. . ." to "We can then state that the changes. . ." Page 9, Line 16: replace ". . .10 (a) & (c). . ." to ". . .10 (a) and (c). . ." *(Done)*
Page 9, Line 21: " Sagres stations . . ." to "Sagres station. . ." *(Done)*
Page 10, Line 3: "Several specific days . . ." to "Two specific day. . ." *(Done)*
Page 19, Table 1 and 2: Part of the table at the right side is missing. *(Revised)*
Recommendation for the figures legends: Include the period over which mean field AOD and AE are calculated and avoid abbreviations such as S.L (significant level) *(Revised)*

**Anonymous Referee #2.**

Rev. #2: The objective of this paper is to quantify the aerosol radiative feedback for the Iberian Peninsula for some pollution episodes. [...] The subject of this study is relevant for publication in ACP. It is crucial to accurately estimate feedback of aerosols from different sources to radiation budget over the region.

*A: As for the anonymous referee #1, we would like to thank to anonymous referee #2 for their valuable comments in the interactive comment on "Assessment of the radiative effects of aerosols in an on-line coupled model over the Iberian Peninsula" by Laura Palacios-Peña et al. (No. acp-2016-473). The manuscript has been revised after reviewer's comments in order to correct errors and to introduce the reviewer' suggestions for improving the quality of the paper. Please see below our point-by-point replies:*

Rev. #2: The section 2.1 provides limited information about the WRF-Chem model setup used in the study. Which gas chemistry, microphysics etc. options were used in the model?

*A: This point has been clarified in the revised version of the manuscript. "The following physics options were applied for both simulations, including (or not) aerosol radiative feedbacks: Rapid Radiative Transfer Method for Global (RRTMG) longwave and shortwave radiation scheme; the Yonsei University (YSU) PBL scheme, the NOAH land-surface model, the Lin microphysics scheme and the updated version of the Grell-Devenyi scheme with radiative feedbacks. Further description of the physics can be found in Grell et al. (2005). According to chemistry options, the followings were applied: MADE/SORGAM aerosol scheme; the RADM2 gas phase mechanism and the Fast-J photolysis scheme."*

*This description has been introduced in the revised version of the manuscript.*

Rev. #2: Why did the authors choose the SORGAM module? It's well known that the SORGAM drastically underestimates secondary organic aerosol (SOA) concentrations, consequently total aerosol concentrations. There are versions of the MADE aerosol scheme coupled to new SOA schemes in WRF-Chem (e.g. Tuccella et al., 2015).

*A: As the review indicates the SORGAM module underestimates simulated PM2.5 mass, mainly attributable to SOA (Grell at al., 2005; McKeen et al., 2007 and Tuccella et al., 2012). As reported by Tuccella et al., (2012), one of the most probable reasons for OM underestimation is that the RADM2 chemical mechanism (also used in this work) does not include the oxidation of biogenic monoterpenes and has a limited treatment of anthropogenic VOC oxidation (McKeen et al., 2007).*

*We agree with the reviewer that the election of SORGAM may bring underestimation in SOA levels. However, it is really hard to establish the cause of SORGAM's underestimation, especially because the AOD levels are overestimated during the biomass burning episode.*

*The aforementioned authors also point to other causes of PM negative bias. Finally, another potential source of the PM2.5 bias is the simulation of the meteorological fields, as temperature or wind speed. It could be linked to unspeciated PM2.5 due to underestimation of its emissions. So, the a priori selection of a SOA mechanism is hard to establish.*

*Moreover, the election of SORGAM comes conditioned by the participation of our group in EuMetChem Cost Action ES1004 and AQMEII initiative. Our configuration, which uses the MADE/SORGAM aerosols and the RADM2 gas-phase mechanisms, was established within this Cost Action, where other groups used different configurations of SOA (e.g. VBS) so we may have information about the sensitivity of the WRF-Chem model to the election of several*

*physico-chemical options (such as the election of the SOA mechanism). That's the main cause for the election of the SORGAM module.*

*Grell, G. A., Peckham, S. E., Schmitz, R., McKeen, S. A., Frost, G., Skamarock, W. C., and Eder, B.: Fully coupled "online" chemistry within the WRF model, Atmos. Environ., 39, 6957-6975, 2005.*

*McKeen, S., Chung, S. H., Wilczak, J., Grell, G., Djalalova, I., Peckham, S., Gong, W., Bouchet, V., Moffet, R., Tang, Y., Carmichael, G. R., Mathur, R., and Yu, S.: Evaluation of several PM2.5 forecast models using data collected during the ICARTT/NEAQS 2004 field study, J. Geophys. Res., 112, D10S20, doi: 10.1029/2006JD007608., 2007.*

*Tuccella, P., Curci, G., Visconti, G., Bessagnet, B., Menut, L., and Park, R. J.: Modeling of gas and aerosol with WRF/Chem over Europe: Evaluation and sensitivity study, J. Geophys. Res., 117, D03303, doi: 10.1029/2011JD016302, 2012.*

*Tuccella, P., G. Curci, G. A. Grell, G. Visconti, S. Crumeyrolle, A. Schwarzenboeck and A. A. Mensah.: A new chemistry option in WRF-Chem v. 3.4 for the simulation of direct and indirect aerosol effects using VBS: evaluation against IMPACT-EUCAARI data, Geosci. Model Dev., 8(9), 2749-2776, 2015.*

Rev. #2: The authors need to provide more details on how the aerosol-radiation and aerosol-cloud interactions are parameterized in their version of WRF-Chem. These details could help to better interpret the model-observation discrepancies.

*A: This fact is also highlighted by the Anonymous Referee #1, so we refer to the answer above where a detailed description of the aerosol, microphysical and optical modules as well as the previous description of the representation of aerosol-radiation-clouds interactions has been done. This has been included in the revised version of the manuscript.*

Rev. #2: The model was run on 23km resolution. This is a relatively coarse model grid. It doesn't allow simulating land-sea breeze and other mesoscale circulations. Moreover, in such resolution there are more parameterized (by cumulus parameterization) clouds in the model. Since the model doesn't treat aerosol-cloud feedback in cumulus parameterization, the overall ACI effect can't be captured by these model settings.

*A: In spite of the relatively coarse model grid, the model allows the representation of land-sea breezes. The next figure represents the time series of wind on a point in the east coast of the Iberian Peninsula. In this figure we can see the daily cycle due to the land-sea breeze.*

[Figure]

WINDX_C11_ES1 from COST RS 1004 Russian forest fire case, no feedbacks

*According to the treatment of aerosol-cloud feedbacks, as the reviewer indicated and as reported by Archer-Nicholls et al., (2016), WRF-Chem has a limitation to assess aerosol-cloud interactions because the couplings are not computed in convective clouds simulated by the cumulus parameterisation (Chapman et al., 2009; Yang et al., 2011). We are well aware that the limitation of the model but a WRF-Chem state of the art version has been implemented. In spite of this, thanks to the reviewer's comment, we have evaluated the cumulus presence in the episodes study. The next figure shows the mean accumulated convective precipitation as a representation of the cumulus presence. A threshold of 0.25 mm day⁻¹ is considered, being values under this threshold negligible. The figure shows that the highest values, around 5 mm day⁻¹, are found over the north-east of the domain (over the Pyrenees mountains). So, we understand that during both episodes the cumulus presence is limited. Moreover and according to the AOD values, shown in the initial version of the manuscript, the area with the*

*highest values of convective precipitation is not strongly affected by the high aerosol loads study in this work.*

*In spite of this, a comment about the limitation of the model due to the aerosol-cumulus interactions has been done in the revised version of the manuscript.*

*Archer-Nicholls, S., Lowe, D., Schultz, D. M., and McFiggans, G.: Aerosol–radiation–cloud interactions in a regional coupled model: the effects of convective parameterisation and resolution, Atmos. Chem. Phys., 16, 5573-5594, doi:10.5194/acp-16-5573-2016, 2016.*

*Chapman, E. G., Gustafson Jr., W. I., Easter, R. C., Barnard, J. C., Ghan, S. J., Pekour, M. S., and Fast, J. D.: Coupling aerosol-cloud-radiative processes in the WRF-Chem model: Investigating the radiative impact of elevated point sources, Atmos. Chem. Phys., 9, 945-964, doi:10.5194/acp-9-945-2009, 2009.*

*Yang, Q., W. I. Gustafson Jr., Fast, J. D., Wang, H., Easter, R. C., Morrison, H., Lee, Y.-N., Chapman, E. G., Spak, S. N., and Mena-Carrasco, M. A.: Assessing regional scale predictions of aerosols, marine stratocumulus, and their interactions during VOCALS-REx using WRF-Chem, Atmos. Chem. Phys., 11, 11951–11975, doi:10.5194/acp-11-11951-2011, 2011.*

[Figure]

Rev. #2: Another uncertainty stems from using ECMWF analysis fields for the meteorological initial and boundary conditions in the regional WRF-Chem modeling. The ECMWF model assimilates met. observations, which might be already affected by those dust and fire aerosols. Hence, the base WRF-Chem model case implicitly may already include some of the aerosol feedback. I understand that it's hard to set up a "perfect" regional modeling framework to study the aerosol-meteorology interactions, however this issue needs to be mentioned in the paper.

*A: The reviewer is right and this issue has been mentioned in the revised version of the manuscript. Page 7: "At this point, it should be mentioned that the use of ECMWF operational archive for meteorological initial and boundary conditions can produce that some of the aerosol feedback may already take into account in the base case (NRF) because of the model assimilation of meteorological observations of the ECMWF."*

Rev. #2: I don't see much discussions of the simulated ACI effect in the paper. For clarity it'd better to show three model cases - w/o any aerosol feedback, with aerosol feedback on radiation and with aerosol feedback on radiation+clouds, and discuss them more thoroughly.

*A: We appreciate the reviewer's suggestion. However, for the sake of brevity and in order not to increase the length of the manuscript (what would affect readability of the paper) we decided to show only the accumulated effects of NRF versus ARI+ACI cases.*

Rev. #2: Another missing piece in this paper is lack of evaluations of the simulated aerosol concentrations. Thus, it's hard to interpret AOD comparisons given the lack of information about the model's skill to simulate aerosol mass concentrations in dust and smoke plumes.

*A: An evaluation of the simulated aerosol concentrations is presented by Im et al., (2015), where PM simulations for the year 2010 in the context of AQMEII2 are evaluated. The simulations evaluated in the manuscript on revision present the same configuration of the ES1 simulation from Im et al., (2015) and therefore the evaluation results have been mentioned in the revised version of the manuscript citing the work of Im et al. (2015).*

*This point has been clarified in the revised version of the manuscript.*

*Im, U., Bianconi, R., Solazzo, E., Kioutsioukis, I., Badia, A., Balzarini, A., Baró, R., Bellasio, R., Brunner, D., Chemel, C., Curci, G., Denier van der Gon, H., Flemming, J., Forkel, R., Giordano, L., Jiménez-Guerrero, P., Hirtl, M., Hodzic, A., Honzak, L., Jorba, O., Knote, C., Makar, P.A., Manders-Groot, A., Neal, L., Pérez, J.L., Pirovano, G., Pouliot, G., San Jose, R., Savage, N., Schroder, W., Sokhi, R.S., Syrakov, D., Torian, A., Tuccella, P., Wang, K., Werhahn, J., Wolke, R., Zabkar, R., Zhang, Y., Zhang, J., Hogrefe,C. and Galmarini, S.: Evaluation of operational online-coupled regional air quality models over Europe and North America in the context of AQMEII phase2. Part II: particulate matter, Atmos. Environ., 115, 421–441, 2015.*

**Minor comments:**

Rev. #2: Authors use many abbreviations in the text. I suggest adding a table showing all of them in one place.

*A: A table showing all of abbreviations had been included as Apendix in the revised version of the manuscript.*

***Apendix. List of acronyms.***

| | |
|---|---|
| ***ACI*** | *Aerosol-cloud interactions* |
| ***AE*** | *Angström Exponent* |

| | |
|---|---|
| **AERONET** | *AErosol Robotic NETwork* |
| **AOD** | *Aerosol Optical Depth* |
| **ARI** | *Aerosol-radiation interactions* |
| **BSCAT** | *Backscatter* |
| **DB** | *Deep Blue* |
| **DT** | *Dark Target* |
| **EARLINET** | *European Aerosol Research Lidar Network* |
| **ECMWF** | *European Centre for Medium-Range Weather Forecasts* |
| **EuMetChem** | *European framework for online integrated air quality and meteorology modelling* |
| **IFS-MOZART** | *Integrated Forecasting System - Model for ozone and related tracers* |
| **IP** | *Iberian Peninsula* |
| **IPCC** | *Intergovernmental Panel on Climate Change* |
| **IS4FIRES** | *Integrated monitoring and modelling system for wild-land fires* |
| **MACC-II** | *Monitoring Atmospheric Composition and Climate-Interim Implementation* |
| **MAE** | *Mean Absolute Error* |
| **MBE** | *Mean Bias Error* |
| **MEGAN** | *Model of Emissions of Gases and Aerosols from Nature* |
| **MODIS** | *Moderate Resolution Imaging Spectroratiometer* |
| **NRF** | *No radiative feedbacks* |
| **r** | *Correlation Coefficient* |
| **RF** | *Radiative feedbacks* |
| **RRTMG** | *Rapid Radiative Transfer Method for Global* |
| **S.L.** | *Significance Level* |
| **TNO** | *Netherlands Organization for Applied Scientific Research* |
| **YSU PBL** | *Yonsei University Planetary Boundary scheme* |
| **WRF-Chem** | *Weather Research and Forecasting model coupled with Chemistry* |

page 9: correct "values shows" *(Done)*
page 11: correct "fires particles" *(Done)*
References section: The paper by Iacono et al. is entered twice. *(Revised and corrected)*